# Significance of the Śrāvastī Miracles According to Buddhist Texts and Dvāravatī Artefacts

Natchapol Sirisawad

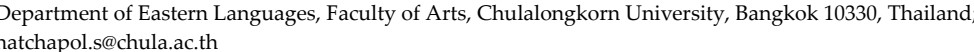

Department of Eastern Languages, Faculty of Arts, Chulalongkorn University, Bangkok 10330, Thailand; natchapol.s@chula.ac.th

**Abstract:** The story of the Śrāvastī miracles is one episode of the Buddha's biography that is depicted in the art forms of Dvāravatī from about the 7th to the 11th centuries CE. The fact that many artefacts were produced—in such variety, over such a long period, and at so many sites—shows the popularity of the scene of the Śrāvastī miracles in the Dvāravatī culture. The objective of this research paper is to analyze the significance of the story of the Śrāvastī miracles that affected the creation of Dvāravatī art in Thailand by examining the textual sources together with the Dvāravatī artefacts. The analysis shows that the stories of the Śrāvastī miracles were significant in various ways, being one of the Buddha's necessary deeds, a principal miracle only performed by the Buddha, a means of converting others to Buddhism, and a key source for the idea of making Buddha images as an act of merit. These significant features may explain the popularity of the Śrāvastī miracle theme in Dvāravatī culture.

**Keywords:** Śrāvastī miracles; Twin Miracle; Buddhist texts; votive tablets; Dvāravatī

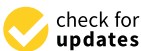



## 1. Introduction

In my recent research on "The Śrāvastī Miracles: Some Relationships between their Literary Sources and Visual Representations in Dvāravatī," the visual representations of 24 artefacts in stone, stucco, and terracotta that were found in central, northeastern, and southern parts of what is present Thailand have been comparatively studied in relation to certain textual sources (Sirisawad 2022). The numerosity of these artefacts demonstrates the predilection for depicting the Śrāvastī miracles in Dvāravatī culture between the 7th and 11th centuries CE and leads us to conclude that this episode from the Buddha's biography held considerable significance at that time. Dvāravatī is seen as the first historic culture in Thailand (Glover 2017, p. 16). According to Brown (1996, p. XXI), Dvāravatī is a culture that encompassed most of present-day Thailand and is associated with extensive art and architectural remains. However, little is known about its history. There are a few Dvāravatī inscriptions, but they are almost exclusively associated with Indian religious texts. Other written information seems to come from some brief references in Chinese records. As such, this research paper then seeks to uncover the reasons for the popularity of the Śrāvastī miracle by examining the textual sources and the Dvāravatī artefacts, and why they have spurred the creation of Dvāravatī art in Thailand.

The narratives concerning the Śrāvastī miracles are numerous. They were widely transmitted and are today preserved in various versions written in Sanskrit, Pāli, Gāndhārī, Tibetan, Chinese, Mongolian, and Thai. Most of the textual narratives used in this study belong to the traditions of the Mūlasarvāstivādins, the Theravādins, and the Dharmaguptakas, in addition to some of an unidentified school affiliation. The Mūlasarvāstivāda version of the narrative is attested in numerous sources; the Sanskrit *Mahāpratihāryasūtra* of the Gilgit manuscripts, an important collection of Buddhist manuscripts from sixth–eighth centuries CE [=MPrS], the Tibetan translation of the *Kṣudrakavastu* (*'Dul ba phran tshegs kyi gzhi*) [=MSV-T], which is attributed to Vidyākaraprabha, Dharmaśrīprabha and dPal 'byor who were active in the early ninth century CE, and the Chinese Translation of the *Kṣudrakavastu*

(T. 1451 根本說一切有部毘奈耶雜事, vol. 24: 329a5–333c14) contributed by Yijing (義淨) in 710 CE [=MSV-C]. These two Tibetan and Chinese parallels in the *Kṣudrakavastu* are part of the Mūlasarvāstivāda *Vinaya*. Other witnesses include the *Prātihāryasūtra*, the 12th narrative of the *Divyāvadāna* [=PrS(Divy)], some citations from the *Mahāprātihāryasūtra* found in Śamathadeva's *Abhidharmakośopāyikāṭīkā* [=Upāyikā-ṭīkā], which is essentially a commentary on the *Abhidharmakośabhāṣya*, and the "*Pratihāryāvadāna*," the 13th century narrative of the *Bodhisattvāvadānakalpalatā* [=Av-klp], which was collected and arranged in verse in the middle of the 11th century CE by the Kashmiri poet Kṣemendra. In the Theravāda versions, the narrative is recorded in the *Yamakapāṭihāriyavatthu*, which is a part of Buddhaghosa's commentary to the *Dhammapada* (*Dhammapadāṭṭhakathā*) [=Dhp-a], as well as in the Pāli *Jātaka* Commentary (*Jātakaṭṭhakathā*) as the *Paccuppannavatthu*, the first part of the *Sarabhamigajātaka* (the *Jātaka* of the Deer) no. 483 [=Ja]. In the Dharmaguptaka and related versions, the stories are preserved in the Chinese translation of the Dharmaguptaka *Vinaya* (T. 1428 四分律, vol. 22: 946b–951c translated by Buddhayaśas (佛陀耶舍) and Zhu Fonian (竺佛念), 408–413 CE), T. 202 (賢愚經, vol. 4: 360c–366a, complied by Huijue 慧覺, 445 CE), and in T. 160 (菩薩本生鬘論, vol. 3: 334c–338b, written by Āryaśūra 聖勇, translated by Probationary Assistant Minister of Court of State Ceremonial 試鴻臚少卿 during the 12th century CE). The other versions of the unidentified school affiliation include T. 211 (法句譬喻經, *Dharmapadāvadānasūtra*, vol. 4: 598c–599c, translated by Faju (法炬) and Fali (法立), third to the forth centuries CE), and T. 193 (佛本行經, vol. 4: 83c28–87a3, translated around 424–453 CE).[1] The earliest literary form of this narrative is a Gāndhārī text composed throughout in *mātrācchandas* that is preserved in a c. first century CE birch-bark manuscript of the Split-Collection of Kharoṣṭhī Manuscripts.[2]

In the first half of this paper, I will comparatively review the textual traditions of various Buddhist schools based on different textual sources regarding the significance of the Śrāvatī Miracles. The Śrāvatī Miracles referred to here are mainly the Twin Miracle (*yamakaprātihārya*), where the Buddha rises to the mid-air and emits both fire and water from his body, and the Great Miracle (*mahāprātihārya*) where the Buddha duplicates himself in multitude. In this section, I will discuss how these miracles are presented to position the Buddha as supreme to leaders of non-Buddhist sects and as one of the necessary deeds of a Buddha to inspire his audience with awe and to subdue his opponent, perhaps as a prelude to the extraordinary expositions of the Buddha that comes later to convert them to Buddhism. In the second part, I will demonstrate how the significance of these miracles discussed earlier is relevant and important in establishing Buddhism in Dvāravatī by using votive tablets as a means. The depictions of these tablets tell us that Buddhism, viewed as a foreign religion, was in the process of being integrated into the Dvāravatī community. What might happen is that the promulgators of Buddhism at that time, supported by the ideology of merit, seem to have found that the Śrāvatī Miracles narrative could serve as an important means to glorify the Buddha and shatter indigenous or other religious belief systems in order to make way for Buddhism.

## 2. The Significance of the Śrāvastī Miracles According to Buddhist Texts and Dvāravatī Artefacts

According to these Buddhist texts (especially those of the Mūlasarvāstivādins and Theravādins) as well as the artistic expressions of the Dvāravatī artefacts, the story of the Śrāvastī miracles has certain significance, and this may well explain its popularity leading to the numerous artistic depictions in the Dvāravatī period. The following are the most obvious.

### 2.1. Śrāvastī Miracles as One of the Buddha's Necessary Deeds

Various Buddhist scriptures indicate that the performance of the miracles at Śrāvastī is one of the actions that a Buddha must accomplish before his nirvāṇa. Exclusively in the Mūlasarvāstivāda versions, it is recorded that the Buddha enumerates five or ten "necessary deeds" (*avaśyakaraṇīya*) that are to be performed by all Buddhas. The types of deeds in

the list vary among various traditions (Sirisawad 2019, p. 216). The shortest thereof is contained in the Kṣudrakavastu which enumerates five deeds (MSV-T: *nges par mdzad pa*; MSV-C: 五事必定須作), namely: (1) all Buddhas must at some point in their final lives inspire others to make vows for Buddhahood, (2) designate a disciple as an heir of the dharma King, (3) convert their parents, (4) put on a miracle display at Śrāvastī, and (5) teach all those beings they are destined to be taught[3] (MSV-T: Sirisawad 2019, p. 106, § 7.4; MSV-C: T. 1451 329c26–330a2). The Chinese *Ekottarikāgama* (T. 125) also mentions five essential deeds; however, these do not correspond to those in the Tibetan and Chinese translations of the *Kṣudrakavastu*[4]. In the *Mahāvastu* (i 51), a previous Buddha named Samitāvin also mentions "five deeds that are necessarily done by the Buddha" (*paṃca ca buddhakāryāṇi avaśyaṃ kartavyāni*). While ten necessary deeds are found in the *Vinayavibhaṅga* (Q1032, vol. 42, 66a3, French transl. Feer 1883, pp. 47–48), the *Bhaiṣajyavastu* of the *Vinayavastu* (Gilgit and the Tibetan versions[5]), the *Prātihāryasūtra* of the *Divyāvadāna* (*daśa-avaśyakaraṇīyāni*), and the *Kaṭhināvadāna* (Degener 1990, pp. 22–23, 43, § 12), which is quite similar to the Tibetan version of the *Bhaiṣajyavastu*.

The Buddha's display of the Great Miracle at Śrāvastī is regarded as a unique and obligatory event in his life, according to the scriptures related to the Mūlasarvāstivāda versions. The display of the Śrāvastī miracle was mentioned as the last *avaśyakaraṇīya* in the *Prātihāryasūtra* of the *Divyāvadāna*, which was probably a deliberate choice intended to emphasise the event. However, this significant event can be performed only by a fully enlightened Buddha and it is never found in any previous birth narrative, as indicated in the Buddha's thought in the *Kṣudrakavastu*: "What the Blessed One thought is that 'Where did the former complete and perfect Buddhas perform the Great Miracle?' and he realized that it was in Śrāvastī" (MSV-T: Sirisawad 2019, p. 107, § 7.5; MSV-C: T. 1451 330a3–4). Table 1 shows how the numbering and order of the list of these necessary deeds varies among individual scriptures.

**Table 1.** List of necessary deeds of a Buddha in various Buddhist scriptures.

| *Avaśyakaraṇīya* | MSV-T | MSV-C | *Ekottarikāgama* | | Mvu | *Vinaya vibhaṅga* | *Bhaiṣajyavastu* | | PrS (Divy) | *Kaṭhināvadāna* |
|---|---|---|---|---|---|---|---|---|---|---|
| | | | T. 125 [a] | T. 125 [b] | | | Gilgit | Tib | | |
| (i) To help sentient beings to engage in the search of ultimate awakening | 1 | 1 | | | | 2 | 2 | 2 | 2 | 2 |
| (ii) To turn the wheel of dharma | | | 1 | 1 | 1 | | | | | |
| (iii) To consecrate as heir apparent a disciple who has accumulated the roots of virtue | 2 | 2 | | | | | | | | |
| (iv) To establish his mother and father in the truth | 3 | 3 | | | | 10 | 9 | 9 | 9 | 9 |
| (v) To preach the dharma to his parents | | | | 2 | | | | | | |
| (vi) To preach the dharma to his father | | | 2 | | 3 | | | | | |
| (vii) To preach the dharma to his mother | | | 3 | | 2 | | | | | |

**Table 1.** *Cont.*

| *Avaśyakaraṇīya* | MSV-T | MSV-C | *Ekottarikāgama* | | Mvu | *Vinaya vibhaṅga* | *Bhaiṣajyavastu* | | PrS (Divy) | *Kaṭhināvadāna* |
| --- | --- | --- | --- | --- | --- | --- | --- | --- | --- | --- |
| | | | T. 125 [a] | T. 125 [b] | | | Gilgit | Tib | | |
| (viii) To display the Great Miracle at Śrāvastī | 4 | 4 | | | | 8 | 7 | 7 | 10 | 7 |
| (ix) To convert all those who should be converted | 5 | 5 [a] | | | 4 | 3 | 3 | 3 | 3 | 3 |
| (x) To lead those people who do not have faith to the ground of faith | | | | 3 | | | | | | |
| (xi) To generate the aspiration for Bodhisattvahood | | | 4 | 4 | | | | | | |
| (xii) To give a prediction to the Bodhisattva/ to prophesize a future Buddha. | | | 5 | 5 | | 1 | 1 | 1 | 1 | 1 |
| (xiii) To exceed the fifth part of his lifetime | | | | | | 4 | | 6 | | 6 |
| (xiv) To exceed three-quarters of the duration of his existence. | | | | | | | 4 | 4 | | |
| (xv) To draw a strict line of (moral) demarcation (between good and evil) [b] | | | | | | 5 | 5 | 5 | 5 | 5 |
| (xvi) To appoint a pair of his disciples as the foremost of all | | | | | | 6 | 6 | 4 | 6 | 4 |
| (xvii) To display his descent from the heaven of the devas to the city of Sāṃkāśya | | | | | | 7 | 8 | 8 | 7 [c] | 8 |
| (xviii) To explain the sequence of his previous actions at the great lake Anavatapta. [d] | | | | | | 9 | 10 | 10 | 8 | 10 |

[a] See note 3; [b] See (Rotman 2008, p. 430, note 588); [c] See Rotman 2008, p. 430, note 589; [d] See (Lamotte 1958, p. 767) (Webb-Boin, transl., Lamotte 1988, pp. 659–60).

According to the Theravādins, the Buddha does not perform the Great Miracle at Śrāvastī but the Twin Miracle. As in the passage from the *Sumaṅgalavilāsinī* (ii 412), a miraculous display of superhuman power (*iddhipāṭihāriya*) known as the Twin Miracle (*yamakapāṭihāriya*) is included in a series of specific episodes in the life of the Buddha which are referred to as miracles (*pāṭihāriya*):

Sv: [As Bodhisattvas (i.e., future Buddhas) in our final birth], we will display miracles (*pāṭihāriya*) that will, among other things, shake the earth, which is bounded

by the circle of ten thousand mountains, when (1) the all-knowing Bodhisattva enters his mother's womb, (2) is born, (3) attains awakening, (4) turns the wheel of dharma, (5) performs the "twin miracle" (*yamakapāṭihāriya*), (6) descends from the realm of the gods, (7) releases his life force, [and] (8) attains cessation (Fiordalis 2008, p. 49).

Some have speculated that the usage of *pāṭihāriya* to denote each of the events listed in the series (birth, awakening, the first preaching, and death) shows an extension of the meaning of the term. The range of its meaning is extended from its original reference to a miraculous display of superhuman power (*iddhipāṭihāriya*), such as the Twin Miracle, which suggests an early and close connection between *iddhi*, "superhuman power," and *pāṭihāriya*, "miraculous or marvelous display,"[6] to other events perceived as wondrous, prodigious, or somehow miraculous for other reasons (Fiordalis 2008, pp. 49–50). The four marvellous and fantastic events that occur when a fully awakened Buddha comes into existence are listed in a series of *Acchariyasuttas* (AN ii 130ff Cf. MN iii 118ff) as follows: (1) descent into the womb, (2) birth, (3) awakening, and (4) the first preaching of the dharma. These specific events in the life of the Buddha were perceived as miracles and were marked by fabulous rays of light as signs of portentous events.

### 2.2. *Śrāvastī Miracles as Principal Miracles Performed Only by the Buddha*

The Buddha performed various kinds of miracles for the benefits of others and, at times, to subdue the pride or hostility of non-Buddhist ascetics. Two prominent miraculous demonstrations in the textual traditions are the "Twin Miracle", or "*yamakaprātihārya*", and the so-called "Great Miracle", or "*mahāprātihārya*".

For the Theravādins, there are no accounts of the performance of the *yamakaprātihārya* by any of the Buddha's disciples or auditors. The Twin Miracle in its aspects of the miracle of fire and water can be performed only by a Buddha. The *Paṭisambhidāmagga* states categorically that the *yamakaprātihārya* is not shared by disciples (*asādhāraṇaṃ sāvakānaṃ*). The *Vimuttimagga* of Upatissa Thera, a manual associated with Abhayagiri school of the Theravaṃsa tradition based in Sīhaladīpa (Ceylon), also states that the Twin Miracle is an attainment of the Enlightened One and not of auditors (Skilling 1997, vol. 2, p. 309). Apart from his display at the foot of the *gaṇḍamba* tree (*gaṇḍambamūla*) in Sāvatthī, according to the *Nidānakathā*, a prelude to the *Jātaka* collection of the Theravādins, the author Buddhaghosa notes that the Buddha performed this miracle right after his awakening, at the end of the first week on the seat of the Great Enlightenment (*mahābodhimaṇḍa*), in order to dispel the doubts of the gods about his attainment. The Buddha performed the same miracle again when he was challenged to a contest of his power by the wanderer Pāṭikaputta (*Pāṭikaputtasamāgama*) at Vesāli, while visiting his relatives (*ñātisamāgama*) at Kapilavatthu (Skilling 1997, vol. 2, pp. 310–11).

Similarly, according to the Mūlasarvāstivādins, the Buddha did not only perform the *yamakaprātihārya* at Śrāvastī, but also at *Gayāśirṣa* for the benefit of one thousand former *jaṭilas*, and then at Kapilavastu for the benefit of the Śākyans as well as Yaśodharā and the women of the harem. However, the Mūlasarvāstivādins also indicate that the Buddha's disciples and the *pratyekabuddhas* (independently awakened ones) also performed this kind of miracle in different places and on different occasions (Skilling 1997, vol.2, pp. 304–7). While the performance of the "miracle of fire and water" in some textual traditions is common to disciples (MSV-T, MSV-C, PrS(Divy)), the independently awakened ones (MSV-T), and Buddhas (MSV-C), the "miracle of the emanation Buddhas," which is called the "Great Miracle" (mahāprātihārya) in the Mūlasarvāstivāda scriptures, can be performed only by a Buddha and is not shared by auditors (see Sirisawad 2019, pp. 245–46).

### 2.3. *Śrāvastī Miracles as a Means of Conversion into Buddhism*

The Buddha's mission and purpose was to lead living beings to freedom from suffering and he became the primary and highest teacher of the dharma (Skilling 2009, p. 76).

The miraculous demonstration offers the possibility to visualize and perceive the dharma. Miracles exemplified in the life of the Buddha are full of stories of the Buddha or his disciples flying through the air, walking on water, shooting out flames from their bodies, causing earthquakes, or visiting the gods (Strong 2013, p. 13). The older Vedic and epic literature, particularly the *Mahābhārata*, and the literature of early ascetic groups such as the Jains had an influence on the conceptual background of South Asian Buddhist discourse on miracles and superhuman powers, as evidenced by many of the Buddhist terms for miracles and superhuman powers (Fiordalis 2008, p. 8; De Notariis 2019, pp. 227–64).

Terms such as *prātihārya* (miracle), *ṛddhi* (supernatural power), *abhijñā* (superknowledge), *uttaramanuṣyadharma* (superhuman phenomenon), *āścaryādbhutadharma* (wonderful and marvelous phenomenon), and *vidyā* (knowledge)[7], are all frequently used to describe superhuman abilities or miraculous events. However, the first, *prātihārya*, or its Pāli equivalent, *pāṭihāriya °hārika, °hera, °hīra*, is perhaps the most common, denoting a variety of miraculous, marvelous, and portentous events (Fiordalis 2008, p. 48). The term *prātihārya* means "extraordinary occurrence, miracle." (BHSD 392) The *Sanskrit-Wörterbuch der buddhistischen Texte aus den Turfan-Funden* gives the meaning of the *prātihārya* as "Wunder, Wundertat" (SWTF iii 229). It is often used to designate a "wonder" or "miracle." The word has been translated as "miracle or miraculous display" (Fiordalis 2008, p. 47). The terms *pāṭihāriya* and *acchariya-abbhuta-dhamma* are sometimes used synonymously to refer to specific sets of events that form something like a cycle of miracles in the last life of the Buddha.

Apart from general meaning, this term can also signify the capacity of his teaching to "remove opposition", which derives from the literal meaning of the verb *paṭiharati*, "to take away"[8] (Bodhi 2017, p. 1392, note 538), consisting of two elements including the prefix, *prati-* (Pāli: *paṭi-*), meaning "towards, near to; against, in opposition to; back, again, in return and so on" (MW 661; PTSD 391), combined with the verb root, $\sqrt{hṛ}$ (Pāli: $\sqrt{har}$), which means "to take away, carry off, seize, to remove, destroy, dispel, frustrate, annihilate, to turn away" (MW 1302; PTSD 727). This meaning agrees with the etymological explanation of *iddhi-pāṭihāriya* in the *Paṭisambhidāmagga* (ii 228–229), as follows:

> Paṭis: Renunciation (*nekkhamma*) succeeds: this is supernatural powers (*iddhi*). It removes (*paṭiharati*) desire (*kāma-cchanda*): this is miracle demonstration (*pāṭihāriya*). Non-malevolence (*abyāpada*) succeeds: this is supernatural powers (*iddhi*). It removes (*paṭiharati*) malevolence (*byāpada*): this is miracle demonstration (*pāṭihāriya*). Sight consciousness (*ālokasaññā*) succeeds: this is supernatural powers (*iddhi*). It removes (*paṭiharati*) stolidity and torpor (*thīna-middha*): this is miracle demonstration (*pāṭihāriya*). Calmness (*avikkhepa*) succeeds: this is supernatural powers (*iddhi*). It removes (*paṭiharati*) distraction (*uddhacca*): this is miracle demonstration (*pāṭihāriya*). The arahant path (*arahattamaggo*) succeeds: this is supernatural powers (*iddhi*). It removes (*paṭiharati*) all defilements (*sabba-kilesa*): this is miracle demonstration (*pāṭihāriya*). Therefore, [it is called] iddhipāṭihāriya. (Ñāṇamoli 1982, p. 396)

Corresponding to the *Paṭisambhidāmagga*, the *Līnatthavaṇṇanā* (i 46–47), the subcommentary of the *Dīghanikāya*, also seems to etymologically explain the meaning of *vividhapāṭihāriyaṃ*:

> Sv-pṭ: Here, as to the etymology of the term *pāṭihāriya*, they speak of *pāṭihāriya* because of taking away opponents (*paṭipakkhaharaṇato*), [that is] because of removing such defilements as lust. But the Blessed One has no opponents such as lust to be taken away. In the case of worldlings too, the spiritual powers occur when the opponents [of their minds] have been destroyed, that is, when their minds are devoid of defilements and possessed of eight excellent qualities. Therefore it is not possible to speak of *pāṭihāriya* in the case, using the expression in relation to them. But the defilements in those who are to be trained by the Blessed One, the Great Compassionate One, are opponents; so if the word *pāṭihāriya* is used

because of the 'taking away of those opponents,' in such a case this is correct usage of the term. Or alternatively: The sectarians are the opponents of the Blessed One's teaching. *Pāṭihāriya* signifies the taking away of them. For they are taken away (*haritā*), removed, by means of psychic powers, mind-reading, and instruction, by taking away their views and by rendering them incapable of expounding their views . . . [9] (Bodhi 2017, p. 1392, note 538).

The *Abhidharmakośabhāṣya* and the *Kevaṭṭasutta* seem to use religious conversion as a basis for connecting the three types of miraculous display. In the *Abhidharmakośabhāṣya*, the super knowledge (*abhijñā*) of supernormal accomplishment—of penetration of the minds of others, and of the exhaustion of all mental intoxicants—is considered as the basis of the three miracles or methods of conversion because it ravishes or converts through three kinds of the miracles: the miraculous display of superhuman powers (*ṛddhi-prātihārya*), the miraculous display of telepathy (*ādeśanā- prātihārya*), and the miraculous display of exposition [in accordance with reality] (*anuśāsanī-pāṭihāriya*) (Abhidh-k-bh 7.47). According to Vasubandhu, the three super knowledges are so-called *prātihārya* because, through them, "the work of conversion (*haraṇa*) is begun and carried out in an intense manner; one ravishes (*harati*) the mind of persons to be converted (*vineyamanas*) from the beginning and very powerfully." Or, because they are used to seize people who hate (*pratihata*) or are indifferent (*madhyastha*) to the true doctrine. He explains that *prātihārya* occurs at the beginning and describes its nature as intense due to the word *prāti-*, which is a combination of two prefixes, *pra + ati*, signifying "the beginning of an action" and "extreme intensity," respectively (Abhidh-k-bh 7.47ab; La Vallée Poussin and Sangpo 2012, p. 2251). These explanations involving argument from the etymology of the word draw a clear connection between miracles and conversion. The three-fold list of miraculous displays are also contained in the *Kevaṭṭasutta* of the *Dīghanikāya* (i 212–214): a miracle demonstration of supernatural powers (*iddhi-pāṭihāriyaṃ*), a miracle demonstration of mind-reading (*ādesanā-pāṭihāriyaṃ*), and a miracle demonstration of admonition (*anusāsanī-pāṭihāriyaṃ*).[10] The Buddha had strongly recommended that monks should abstain from displaying superhuman powers and telepathy, explaining that the true wonder is teaching the dharma. The miracle of instruction thus ultimately subsumes all other types of miracles (Gethin 2011, p. 231). Malalasekera puts forth much the same view, stating that, in this sutta, "the Buddha expresses his hatred of miracles and tells Kevaṭṭa that a greater and better wonder than any or all of them is education in the system of self-training which culminates in Arahantship" (DPPN i 667). Similarly, in the *Abhidharmakośabhāṣya*, Vasubandhu also identifies the miracle of expositions as the best among the miracles because it is never separated from the super knowledge of the exhaustion of fluxes (*āsravakṣaya*) and it confers the effects of salvation and happiness. Whereas the first two miracles can be merely produced by means of supernatural clear knowledge (*vidyā*) called the Gāndhārī *vidyā* (flying through space) and the Īkṣaṇikā *vidyā* (reading another's thoughts) (Abhidh-k-bh 7.47; La Vallée Poussin and Sangpo 2012, p. 2251).

Some might argue that the miracles of superhuman powers and miracles of mind-reading are ineffective tools for conversion and not effective in bringing people into the Buddhist fold. This could be the case; however, all of this is meant to set up the third *prātihārya* as the only miraculous practice approved by the Buddha. That is the miracle of instruction in the dharma, and no magical charm can substitute, nor is any confusion with non-Buddhist heretics possible (Strong 2013, pp. 25–27). The last miracle demonstration is extolled as the supreme wonder, the other two miracles are also necessary as means of conversion, although "they are capable only of captivating another's thoughts for a short time and do not produce important effects" (Abhidh-k-bh 7.47; La Vallée Poussin and Sangpo 2012, p. 2251). Nevertheless, the achievement of higher or direct levels of knowledge (*abhijñā*) in the dharma, is the accomplishment realized through meditation that leads to the facility of controlling the mind which in turn becomes the basis for the different miraculous manifestations. As a result, we found that the Buddha does something utterly marvelous, unprecedented, and unbelievable, which defies explanation. He displays his superhuman

powers, teaches the dharma, and establishes his preeminence, and people respond with awe and devotion, some with fear, trepidation and even cynicism. It is a prototype in that it provides a good basis for evaluating the Buddhist miracle story as a kind or type of story (Fiordalis 2014, p. 2). In one of the versions of the Śrāvastī miracles, the Dharmaguptaka *Vinaya* says that the miracles performed by the Buddha on the 11th, 12th, and 13th days of this enlightenment (?) are called "three ways of teaching (or miracles)", namely, (1) supernatural power, (2) mind-reading, and (3) preaching. The phrasing of the "three ways of teaching (or miracles)" almost correspond to the explanation in another part of the Dharmaguptaka *Vinaya*[11] (Sirisawad 2019, pp. 261–62). This means the three miracles enable the Buddha's transformative power to overcome doubts when proselytizing unbelievers.[12] In the narratives of the Śrāvastī miracles, the Buddha instructed and converted new followers by means of these three miracles in the same way that previous Buddhas had.

2.3.1. The Miracle of Supernormal Accomplishment

The miraculous demonstration of flying in the air and reading others' minds (*mahā-gandhāravijjā*) was a great ability of the Buddha but it was not considered unique; therefore, in order to convince nonbelievers, a clear differentiation of the uniqueness of the Buddha's powers was required. The supernatural power of the Buddha is expressed in the Buddhist hierarchy of miracles, wherein the Buddha stands at the highest level while his followers or Māra can only accomplish minor magical displays like causing fire, light, rain, and lighting (?). The magical art of Gāndhārī could be practiced through different means, but the Buddha's display of the emission of fire and water from the body, or "Twin Miracle", and the so-called "Great Miracle," or body multiplication, were miraculous feats, both, as in the Theravāda tradition, only Buddhas can perform.

Sirisawad have shown elsewhere (2019, pp. 292–93), the Buddha displays various kinds of miracles at Śrāvastī; including, the Buddha making a (mango) tree grow instantly from a planted seed (Dhp-a, Ja, T. 1428, T. 202, T. 160); his levitating into the air and taking on different postures—seated, standing, walking, lying down, and moving about in different directions (MSV-T, MSV-C, PrS(Divy), Dhp-a, T. 211?); his emission of the light (MSV-T, MSV-C, PrS(Divy), Av-klp, Dhp-a, T. 211, T. 193); his simultaneous emission of fire and water from various parts of his body (or the "Twin Miracle") (MSV-T, MSV-C, PrS(Divy), Dhp-a, Ja, T. 211, T. 193); his magical fashion of duplication—a living image of himself (*buddhanirmita*)—with whom he dialogues on matters of the dharma (PrS(Divy), Dhp-a); and finally, the creating and projecting replicas of his own Buddha body, filling the whole sky with them, up to the highest heaven (or the "Great Miracle", *mahāprātihārya*) (MSV-T, MSV-C, PrS(Divy), *Upāyikā-ṭīkā*, T. 1428?, T. 202?, T. 160?, T. 193). The miraculous element found in the Gāndhārī textual tradition is the Buddha's luminosity and levitation. He appears in a multitude (*koṭidhā*) of copies against the sky and certainly emits light in all directions (Falk and Steinbrückner 2020, pp. 35–36, B26). The visualization in this older Gandhāran narrative probably presents the trace of the one "Twin" (*yamaka*) Miracle. The display simply made visible the faculties that the Buddha was always meant to possess (Falk and Steinbrückner 2020, p. 38).

The most unique of the Śrāvastī miracles, is the manifestation of a duplicate Buddha (PrS(Divy), Dhp-a) or magically created forms of the Buddhas (MSV-T, MSV-C, PrS(Divy), *Upāyikā-ṭīkā*) rising into the sky and answering questions of himself, concerns the dharma. The Buddha teaching the Law while in the sky is represented in Buddhism as the strongest materialisation of his teaching. "The point of the sky-lectures is conversion, as it is one way in which the Buddha awes his audience on the ground and convinces them of the authority of the Law" (Brown 2011, p. 23).

The textual narratives can be classified by virtue of the element of the miracle display. The prominence of the "Twin Miracle" is emphasised in the Theravāda versions (Dhp-a, Ja), because only the Buddha can perform it, whereas it appears as one of the preliminary miracles of the Buddha, which can be performed by any enlightened being in the Mūlasarvāstivāda versions (MSV-T, MSV-C, PrS(Divy)). Two Chinese versions (T. 211,

T.193) also mention this kind of miracle. Another type of miracle called the "Great Miracle", which represents a more advanced stage than the Theravāda versions, is added in the formulation of the textual tradition in the Mūlasarvāstivāda (MSV-T, MSV-C, PrS(Divy), *Upāyikā-ṭīkā*). The Dharmaguptaka and related versions (T. 1428, T. 202, T. 160) do not formulate the miracles at Śrāvastī in the same terms as the Mūlasarvāstivāda tradition by describing the Śrāvastī miracles as taking place over many days. There is a possibility that the "Great Miracle" was limited to the textual traditions of the Mūlasarvāstivāda. The Theravāda version (Dhp-a) seems to introduce a new element, the creation of a duplicate Buddha (*nimmita*), for which only the *Prātihāryasūtra* of the *Divyāvadāna* has a parallel.

The artefacts of the Śrāvastī miracles found in the Dvāravatī period illustrate important narrative elements of miraculous demonstration from various Buddhist traditions, including the miracle tree and the multiplication of the Buddha which comprise of the Twin Miracle, the Great Miracle, the creation of the duplicated Buddha, and displaying a miracle akin to experience in the fourth absorption. Some artefacts combine the narrative elements of both the Theravādins and Mūlasarvāstivādins, while another group, besides depicting the account of both the Theravādins, and the Mūlasarvāstivādins, exhibits influence from probably the Dharmaguptakas. Certain artefects show a narrative motif of the thousand-petalled lotus throne supported by *nāgas* (mystical serpents) as part of the Great Miracle recounted in the Mūlasarvāstivāda tradition (Sirisawad 2022, pp. 19–20).

### 2.3.2. The Miracle of Mind-Reading

In the Mūlasarvāstivāda versions (MPrS, MSV-T, MSV-C, PrS(Divy)), just prior to the Buddha's performance of the Great Miracle at Śrāvastī, the narrative is interrupted by the tale of King Prasenajit's brother, Prince Kāla. The story of Prince Kāla is one of the preliminary episodes leading up to the performance of the Buddha's Great Miracle at Śrāvastī. Prince Kāla, a younger brother of King Prasenajit, suffered the punishment of having his limbs cut off. In the Tibetan and Chinese translations of the *Kṣudrakavastu*, the suffering of Prince Kāla was reported to the Buddha by Ānanda. While Ānanda was wandering for alms, he reached the place where Prince Kāla was. His relatives asked Ānanda to speak a word of truth in order to restore his body but Ānanda had to ask the Buddha. Then Ānanda went back to Śrāvastī and explained the situation to the Buddha in order that he receive some advice (MSV-T: Sirisawad 2019, pp. 121–22, § 9.4–9.5; MSV-C: T. 1451: 330b26–c4). In the *Prātihāryasūtra* of the *Divyāvadāna*, the Buddha exhibited the miracle of mind-reading by making clear that he already knew of the incident from afar without being informed, and that there is nothing unknown to him. He thereupon ordered Ānanda to perform an act of truth for Prince Kāla (PrS(Divy) 154.15–16).

### 2.3.3. The Miracle of Exposition

What causes a group of nonbelievers to be converted to Buddhism is the preaching of the Buddha, and thus such teachings of the dharma for the sake of all living beings are to be found in numerous textual episodes. The miracles performed by the Buddha were designed to awe and impress the viewers or listeners in order that they are convinced of his teachings as the direct product of his personal enlightenment and without any intermediary between him and the dharma. They were also displayed to affect the realization of Buddhist metaphysical conceptions concerning the nature of reality in connection with the power of meditation. All versions are presented in prose other than those of the Mūlasarvāstivādins which are in verse.

a.   Prose sermons

The Buddha seems not to preach the dharma during or after his miraculous demonstration according to the surviving Kharoṣṭhī Manuscripts. The Buddha speaks about the dharma after the arrival of 500 arhats: " . . . the dharma was made to shine by the dharma words. He became even more splendid by these (dharma-words), like the moon by the groups of stars." Then, he gives advice with a soft voice, "Be without temper, be pure, and opposed to intoxication", to enlighten the King of Kosala, who then instructs the heretics

(Falk and Steinbrückner 2020, pp. 33–34, B17–20). After the Buddha demonstrated the miracles, in one of the Mūlasarvāstivādin versions, the Buddha preaches "a discourse on the dharma that penetrates the four noble truths" (*caturāryasatyasaṃprativedhakī*) to the audience at the end (PrS(Divy) 166.13–14). After the Buddha's miraculous display, in the *Bodhisattvāvadānakalpalatā*, the poet adds a teaching, lacking in the Mūlasarvāstivāda versions, associated with the idea of impermanence (Av-klp 13.57). In the Dharmaguptaka *Vinaya,* the Buddha teaches various dharmas to many people; unspecified dharma, taught in various forms, delights a rich householder who had invited the Buddha to a meal (T. 1428: 947b21–23). The Buddha teaches the dharma in innumerable modes over a period of a 15-day miracle. Some specific topics of Buddhist doctrine are dealt with on specific days of the miracle display. For instance, on the fifth day, the dharma of unhindered penetration is produced as sounds from a stream that flowed out from the four sides of the lakes: "*Every act is transient, painful and empty. Every existence has no self. Nirvāṇa is its cessation and extinction*" (T. 1428: 949b8–9). On the twelfth day, the Blessed One explains the dharma concerning mind and thought for the assembly: "This should be thought. This should not be thought. This should be reflected on. This should not be reflected on. This should be cut off. This should be practiced" (T. 1428: 950a4–6). On the 13th day, the Blessed One taught the people that everything is burning (T. 1428: 950a13–14; Rhi 1991, p. 235):

In T. 202, the teaching of the dharma on the fourth day is produced by the sound of the streams which aurally relate (1) the five faculties and five powers (五根五力), (2) the seven [factors of] enlightenment (七覺), (3) the eightfold path (八道), (4) the three insights and six supernatural powers (三明六通), (5) the six *pāramitās* (六度), (5) the four immeasurable [minds] (四等), and (6) great mercy and great pity (大慈大悲) (T 202: 362c12–13). In T.160, the parallel version of T 202, the sound of the streams teaches the eight ways of emancipation (八解脱) and all the *pāramitās* (諸波羅蜜) on the fourth day of the miracles demonstration. It is said that all those who hear this acquire great enlightenment in their minds (T. 160: 336a13–14). According to T. 193, the Buddha shows [the miracles] so that all living beings cherish good minds. He teaches that the three realms are under the rule of the three principles: "These three realms (三界) are impermanent (無常) and nonsubstantial (無堅). There is no-self (無我). [Existence] is suffering and emptiness (苦空). *Nirvāṇa* (滅), unconditioned (無爲), brings peace (安)" (T. 193: 86c13–17).

b.　Verse sermons

Apart from prose sermons, there are four verse sermons quoted in Mūlasarvāstivāda versions. The parallels of three out of the four verses are found in the Tibetan and Chinese translations of the *Kṣudrakavastu*, the *Prātihāryasūtra* of the *Divyāvadāna*, and the *Upāyikā-ṭīkā*, while one verse occurs only in the *Prātihāryasūtra*. These verses crop up in both Sarvāstivāda and Mūlasarvāstivāda literature as well in several combinations elsewhere. Notably they cannot be found in the textual tradition of the Theravādins that relate to these Śrāvastī miracles.

(1) The first verse sermon (*ārabhadhvaṃ niṣkrāmata*) is comprised of two padas (MSV-T, MSV-C, PrS(Divy), *Upāyikā-ṭīkā*) spoken by the Buddha after the deities and other living beings rejoice in the Buddha's Great Miracle (MSV-T, MSV-C) or after the miracles performed by the magically created forms of the Buddhas (PrS(Divy), *Upāyikā-ṭīkā*). The Buddha recites these verses in order that those he would train are first converted; this reason is not mentioned in either the *Upāyikā-ṭīkā* or the *Prātihāryasūtra* (Sirisawad 2019, p. 255).

The theme of these verses concerns heedfulness (*apramāda*) and how to be rid of suffering by abandoning the circle of existence, as in my explanation that "The first verse is an exhortation to act, to put the teaching of the Buddhas into practice. The second verse highlights the efficacy of heedfulness, which is one of the essential, indeed the quintessential, factors of spiritual practice" (Sirisawad 2019, p. 255). The Buddha even reinforces this same message on his deathbed in the famous exhortation: "All compounded things are bound to cease accomplish your aim through heedfulness! (*vayadhammā saṅkhārā appamādena sampādetha*) (DN ii 120).[13]

(2) The second verse sermon (*tāvad abhāsate kṛmir*), spoken simultaneously by the magically created Buddhas (MSV-T, MSV-C) or by the Buddha himself (PrS(Divy), *Upāyikā-ṭīkā*) in different sequences of the narratives, is made up of two verses (see Sirisawad 2019, p. 255). They liken the *tīrthikas* or the non-Buddhist ascetics to fireflies whose light is incomparable to that of the sun, i.e., the Buddha.[14]

(3) The third verse sermon (*bahavaḥ śaraṇam yānti*), spoken by the Buddha, comprises five *padas*. The Buddha gives instruction on taking refuge in the Three Jewels (see Sirisawad 2019, p. 265). These refuge verses are also mentioned in the *Bodhisattvāvadāna-kalpalatā* as an important component of the Buddha's teaching (Av-klp 13.58–59). One of them (the fourth *pada*) gained independent popularity as the epitome of the Buddha's teaching.

Apart from the *Prātihāryasūtra*, the five verses occur in the *Dhvajāgra-mahāsūtra* (Q959, vol. 38, 293a1) of the (Mūla)sarvāstivāda tradition.[15] Compared with other Tibetan versions of the *Udānavarga* (Q5600, vol. 119, 33a6), the *Udānavargavivaraṇa* (UvViv ii 759), and the *Abhidharmakośabhāṣya* (Q5591, vol. 115, 213a1), these verses seem slightly different. In addition to the Tibetan versions of the Mūlasarvāstivāda tradition, they are found in both Sanskrit and Pāli. In Sanskrit the verses occur in the *Paśyavarga* of the *Udānavarga* (Uv xxvii: 31–35), the *Abhidharmakośabhāṣya* (Abhidh-k-bh 4.32), and the *Vibhāṣāprabhāvṛtti*, the commentary on the *Abhidharmadīpa* (Jaini 1977, p. 127), wherein all five *padas* are used to illustrate "the meaning of refuge" (*śaraṇārtha*). The first verse of the group is quoted in the *Sphuṭārthā Śrīghanācārasaṅgrahaṭīkā* (Singh 1983, p. 49) for the same purpose. Four of them, with the omission of the fourth, occur together in the *Śaraṇavarga* (13: 1–4) of the Patna *Dhammapada*, and are cited in another translation of the *Vibhāṣā* and in a Sanskrit yoga manual from Central Asia (Schlingloff 1964, p. 184). It is evident that the verses are regarded an early and authoritative group concerning the subject of refuge (Skilling 1997, vol. 2, p. 464). In Pāli works, the teaching occurs in the *Buddhavagga* of the *Dhammapada* (Dhp 14: 188–192). In the *Saṃyuttanikāya* (SN ii 185) and the *Itivuttaka* (It 17–18), it is found within a different group of verses, identical in both texts, attributed to the Buddha. Here it is introduced by the expression, "When one sees the truths of the noble ones with true wisdom" (*yato ca ariyasaccāni sammappaññāya passati*). They occur five times in the Theragāthā (verse 1259) and Therīgāthā (verses 186, 193, 310, 321). Thus, it is a set formula summarizing the teaching of the Buddhas (Skilling 1991, p. 241).

(4) The fourth verse sermon spoken by the Buddha comprises two *padas* (Table 2). This verse occurs in the conclusion of the story in the *Prātihāryasūtra* of the *Divyāvadāna*. "The first verse is an exhortation to take refuge and to serve the Buddha, as a result of which they will obtain *nirvāṇa*. The second verse emphasizes the first in stating that even one who offers a little service to the Buddha will also attain the eternal state" (Sirisawad 2019, p. 265). These verses are not found in MSV-T, MSV-C, or the *Upāyikā-ṭīkā*. Both verses have a parallel in other parts of the *Upāyikā-ṭīkā* (Q5595, vol. 118, 103a3–5). Only the second verse occurs in the Sanskrit version of the *Abhidharmakośabhāṣya* (Abhidh-k-bh 7.34), and the Tibetan translation of Daśabalaśrīmitra's *Saṃskṛtāsaṃskṛtaviniścaya* (*'dus byas dang 'dus ma byas rnam par nges pa*) (Q5865, vol. 146, Nyo 268a4–5) and Vasubhandhu's *Gāthāsaṃgraha-śāstranāma* (*bstan bcos tshigs su bcad pa bsdus pa*) (Q5603, vol. 119, 250a6).

These verse sermons (1–4) are especially popular within the Mūlasarvāstivāda tradition and arise in several different contexts in their literature. They were used frequently to exemplify the teaching of the Buddha, as Skilling states: "many early teachings were transmitted in verse (*gāthā*). Different schools may share the same verse, but use them differently, and place them in different contexts" (Skilling 1999a, p. 441).

**Table 2.** The fourth verse sermon that was spoken by the Buddha and its parallels in other texts.

| PrS(Divy) 166.24–27 | Upāyikā-ṭīkā | Abhidharma-kośabhāṣya | Saṃskṛtāsaṃskṛta-viniścaya | Gāthāsaṃgraha |
|---|---|---|---|---|
| *(1) dhanyās te puruṣā loke ye buddhaṃ śaraṇaṃ gatāḥ \| nirvṛtiṃ te gamiṣyanti buddhakārakṛtau[16] janāḥ \|\|* | *gang zhig 'jig rten sangs rgyas la \|\| skyabs song skyes bu de mchog ste \|\| sangs rgyas bya ba byas skye bo \|\| mya ngan 'das par 'gro bar 'gyur \|\|* | | | |
| *(2) ye 'lpān api jine kārān kariṣyanti vināyake \| vicitraṃ svargam āgamya te lapsyante 'mṛtaṃ padam \|\|* | *gang zhig rgyal ba rnams 'dren la \|\| bya ba cung zad byed gyur ba \|\| de dag mtho ris sna tshogs dag \|\| bgrod nas bdud rtsi go 'phang thob \|\|* | *ye 'nyān[17] api jine kārān kariṣyanti vināyake \| vicitraṃ svargam āgamya te lapsyante 'mṛtaṃ padam \|\|* | *gang zhig rgyal ba rnam 'dren la \|\| mchod ba chung ba 'ang byed 'gyur ba \|\| bde 'gro sna tshogs bgrod nas ni \|\| bdud rtsi go 'phang thob par 'gyur \|\|* | *gang ngag rgyal ba rnams 'dren la \|\| byed pa chung ngu 'ang byed 'gyur ba \|\| de dag mtho ris sna tshogs pa \|\| bgrod na 'chi med gnas thob pa \|\|* |

### 2.3.4. Effects

The story of the Śrāvastī miracles displays dharma transmission event or a sky-lecture for the multitude of living beings. In the texts, the succession of events distinguishing the performance of the miracles followed by the preaching the dharma at the end of the event. Both the miraculous demonstration and the dharma teaching afterwards exert an effect on those who respectively see and hear them.

a.   The Effect of Seeing the Miracle Display of the Buddha

The effect of seeing the miraculous demonstration of the Buddha is not clearly mentioned in the textual traditions of the Mūlasarvāstivādins and Theravādins. However, in those of other schools, people obtain different kinds of effects. According to the Tibetan prose rendering of the *Pratihāryāvadāna* of Kṣemendra's *Bodhisattvāvadānakalpalatā* (see Sirisawad 2019, p. 193, note 3), some obtain the various fruits of *samādhi*: "the people in the assembly all obtained the various fruits of *samādhi*, each in accord with the state of their consciousness" (Black 1997, p. 68). Some multiply their faith and devotion as mentioned in the *Sūtra of the Wise and the Foolish* (T. 202: 363a13–14).

b.   The Effect of Hearing the Preaching of the Buddha

In the Mūlasarvāstivāda versions of the story of Prince Kāla, the prince was led to the Buddha and, after having been taught the Dharma by the Buddha according to his thought, tendency, disposition, and nature, he attains a clear view of the truth (MSV-T), the stage of a non-returner (MSV-T: *phyir mi 'ong pa*; MSV-C: 證不還果), and supernatural power (MSV-T: *rdzu 'phrul*; MSV-C: 神通) (MSV-T: Sirisawad 2019, pp. 125–127, § 9.8–9.9; MSV-C: T. 1451: 330c21–25). Although in the *Prātihāryasūtra* of the *Divyāvadāna*, the Buddha did not teach Prince Kāla the dharma directly, he also achieved the state of non-returner (*anāgāmiphala*) and acquired supernatural power (*ṛddhi*) due to the awe (*saṃvega*) he felt following the miraculous event enacted by the power (*anubhāva*) of the Buddha and the divine power of the devas: "Such is the Buddha's innate power and the divine power of deities. Deeply moved, Prince Kāla directly experienced the reward of the nonreturner and acquired magical powers" (PrS(Divy) 155.8–11; Rotman 2008, p. 270).

At the end of the story, following the Buddha's teaching, people in the assembly obtain different effects. In MSV-T, "many hundreds of sentient beings who have heard will attain the great extraordinary thing" (MSV-T: Sirisawad 2019, pp. 178–179, § 16.3). In MSV-C, "innumerable hundreds of thousands of people had a superior understanding"

(MSV-C: T. 1451: 333a12–13; Rhi 1991, p. 285). The *Divyāvadāna* states that "many hundreds and thousands of beings [accepted] the taking of the refuges as well as the precepts" (PrS(Divy) 166.15–20; Rotman, 2008, p. 286). Precisely what the members of the assembly attain after they listen the Buddha's teaching differs slightly between the texts: the heat stages (PrS(Divy)), the summit stages (PrS(Divy)), the tolerance stages (PrS(Divy)), the highest worldly dharma stages (PrS(Divy))[18], the stage of stream-enterer (MSV-T, MSV-C, PrS(Divy)), the stage of once-returning (MSV-T, MSV-C, PrS(Divy)), the stage of non-returner (MSV-T, MSV-C, PrS(Divy)), the stage of the Arhatship (MSV-T, MSV-C, PrS(Divy)), an aspiration for the enlightenment of Śrāvakas (MSV-T, MSV-C, PrS(Divy)), for the enlightenment of Pratyekabuddhas (MSV-T, MSV-C, PrS(Divy)), for the unsurpassed, complete and perfect awakening (MSV-T), or for supreme wisdom (MSV-C). At the end they all take the triple refuge (MSV-T, MSV-C, PrS(Divy)). A follower of the Buddha is termed an "auditor" (*śrāvaka*), thereby signaling the importance of listening to the Buddha's dharma in early Buddhism. In the *Divyāvadāna*, however, devotees are enjoined to look, as much as to hear, for visual practices are represented as the primary means of cultivating faith (Rotman 2008, p. 448).

In the Dharmaguptaka and related versions, all the people who see the Buddha display the miracles are filled with joy and desire to escape the circle of existence. Then the Buddha teaches the dharma in numerous ways over the 15-day period of the miracles and the people obtain different effects on each of the days. On the eighth, in T. 202, the Buddha displays the essence of dharma to nine hundred thousand followers of the six *tīrthikas*; they remove the defilements, obtain enlightenment, and became arhats (T. 202: 363a10–11). In T. 1428, the people attain the same things; including, the removal of the dust and defilement of the world (遠塵離垢) and obtaining the pure dharma-eye (得法眼淨) (T. 1428: 949a5–951c20).

One of the Chinese versions states that after the Buddha preaches the dharma associated with the three principles, his voice was heard in three thousand worlds and hundreds of million living beings produced the intention for great enlightenment (大道意). Innumerable others give rise to the aspiration for the vehicle of *śrāvakas* (緣覺乘). Many hundred million of other living beings enter upon the course of the (Buddha-)Path (道迹). Additionally, numerous *tīrthikas* abandoned their mistaken views (邪見) (T. 193: 86c19–21). In the Theravāda tradition, the dharma of the Buddha causes faith in multitudes of people: "The Master, having for the confounding of the schismatics performed a two-fold miracle passing marvelous among his disciples, caused faith to spring up in multitudes, then arose and, sitting in the Buddha's seat, declared the Law. Twenty crores of beings drank of the waters of life" (J iv 265; Cowell 1990, vol. 4, p. 168). In this passage, a great multitude of living beings numbering two hundred million obtained the understanding of the truth (*dhammābhisamaya*) (Dhp-a iii 215–216).

In the narratives of the Mūlasarvāstivāda versions, the verse forms of the dharma preached by the Buddha are too long to be inscribed on a Buddha image or a miniature clay tablet or *phra phim* (in Thai พระพิมพ์) which is often enshrined in a larger *stūpa*. Instead of long stanza of the Buddha's preaching, most ancient votive tablets bear the central Buddhist creed, *ye dharmā* verses and were frequently "hidden" in small scripts on top, at the side, at the base, or on the back of the artefacts, which always carry the same in meaning whether be it in Sanskrit or in Pāli. They are written with the Nagari characters of Northern India, the southern Brāhmī or with those scripts of Southeast Asia. This is one of the significant teachings of the Four Noble Truths of Buddhism which corresponds to the Buddha's preaching of "a discourse on the dharma that penetrates the four noble truths" (*caturāryasatyasaṃprativedhakī*) at the end of the *Prātihāryasūtra* of the *Divyāvadāna*. The *ye dhammā gāthā* could be understood as an abbreviated version of the *paṭiccasamuppāda*, which is intimately related to the Four Truths preached by the Buddha at Sarnath (Revire 2014, p. 260).

The *ye dhammā* stanza, which archaeologists and art historians have found on clay tablets in the central northeastern and southern part of Thailand, are roughly dated to the

seventh–eighth centuries CE. (Cœdès 1926, p. 5; Revire 2014, p. 247). The summary of the teaching through the *ye dhammā* stanza is said to be articulated by Assaji, one of the first disciples, in the name of the teacher, the Great Śramaṇa, at the conversion of Upatissa, who in turn transmits it to his companion Kolita. It causes a transformation in these two seekers who will later become the foremost pair of disciples, Sāriputta and Mogallāna.[19] According to Skilling's framework of textual classification, they fall into two distinct groups, depending on whether the first words of the second line *tesaṃ*, as in the canonical version, or *yesaṃ*, a regional peculiarity only attested so far in central Thailand (Skilling 1999b [2542 BE], pp. 180–84; 2003, pp. 273–87).

(1)    *ye dhammā hetuppabhavā*
(2)    *tesaṃ hetuṃ tathāgato āha*
(3)    *tesañ ca yo nirodho*
(4)    *evaṃvādī mahāsamaṇo ti* (Vin i 40).

Those things which proceed from a cause, of these the truth-finder [i.e., Tathāgata] has told the cause, Additionally, that which is their stopping—the great recluse has such a doctrine (Horner 1951, p. 54).

A significant number of moulded clay tablets bearing the *ye dhammā gāthā* in Pāli have been found in central Thailand. They are mainly of two types, both depicting the Śrāvastī miracles in rectangular plaques. The usage of Pāli indicates its intended use in a Theravāda devotional context. The first type (enthroned Buddha, mango tree, and multiplied Buddhas) has predominantly been found in Nakhon Pathom province at Wat Phra Men (Figure 1) and Sanamchandra Palace.[20] On these tablets, the verse is directly stamped along the base, below the main scene, and is made from the same mould (Revire 2014, p. 258). It reads as follows in two lines:

(1) *ye dhammā hetupprabhavā* [21]

(2) *tesaṃ hetuṃ tathāgato āha*

(Dupont 1959, p. 49 as cited in Revire 2014, p. 258).

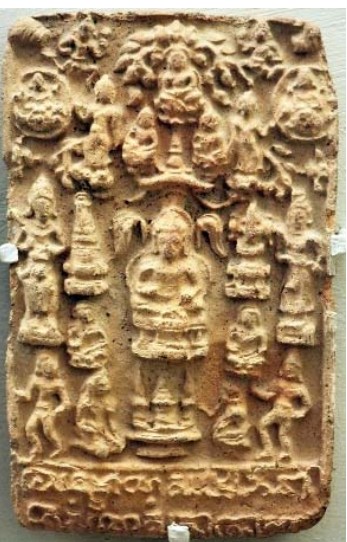

**Figure 1.** The Śrāvastī miracles (enthroned Buddha, mango tree and multiplied Buddhas) with *ye dhammā* formula. Ca. 8th–10th centuries CE. Wat Phra Men, Nakhon Pathom province. Moulded tablet, clay; h. 14 × w. 8 cm. National Museum, Bangkok (DV 6-2). Source: author, 2016.

Only the first two *pādas* of the verse are inscribed on the tablets, and the last two are intentionally omitted (Figure 1). Revire (2014, p. 258) suggests that the first *pādas* read as *hetupprabhavā* instead of *hetuppabhavā*, the standard form of the Pāli, meaning that the inscription exhibits traces of Sanskritisation. The reading is not certain, even though the verse apparently falls in the *tesaṃ* group and the word *tathāgato* was engraved "in a more

condensed way almost as a monogram to engrave all signs on the material available and it is possible that we are dealing here with an abridged version of the stanza".

The entire *ye dhammā* verse is found in the second type of tablets (Buddha meditating beneath a blooming tree and the *nāga*), which are mainly found in Ratchaburi province[22]. On the reverse side of these tablets, the *gāthā* is hard to read since it was incised in cursive handwriting, probably made while the clay was still wet (Figures 2b, 3b and 4b). The inscription of votive tablets from Ratchaburi province occupies four lines (Figures 2b and 3b). The scribe engraved the third line of the verses, the *akṣaras niro-*, until the edge of the votive tablet. Because there is no more space the scribe continued to engrave the *akṣara dh*o beneath *ro* (Figure 2b).

(1)   *ye dhammā hetupprabhavā*
(2)   *yesaṃ hetuṃ tathāgato*
(3)   *āha tesañ ca yo niro-*
       *–dho*
(4)   *evaṃvādī mahāsamano*

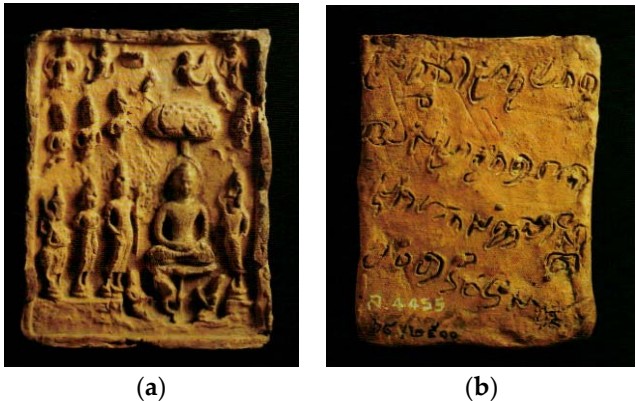

(a)                              (b)

**Figure 2.** The Śrāvastī miracles (Buddha meditating beneath a blooming tree and the nāga) with *ye dhammā* formula. Ca. the first half of the 8th century CE. Ratchaburi province. Moulded tablet, clay; h. 14 × w. 11 cm. U-Thong National Museum (No. 64/2510). (**a**) Font Side; (**b**) back Side. After Bhumadhon and Phongpanit (2015 [2558 BE], fig. 11–69).

(1)   *ye dhammā hetupprabhavā*
(2)   *yesaṃ hetuṃ tathāgato āha*
(3)   *tesañ ca yo nirodho*
(4)   *evaṃvādī mahāsamano*

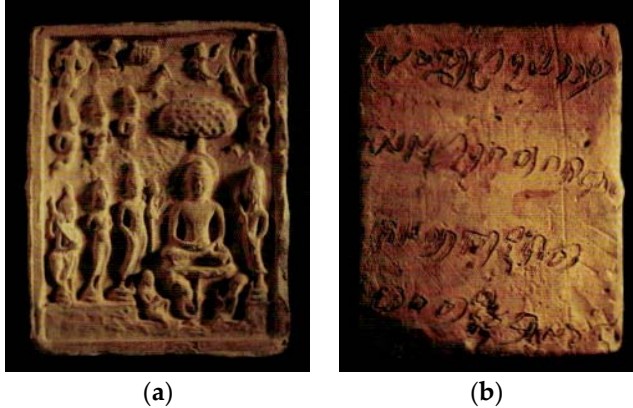

(a)                              (b)

**Figure 3.** The Śrāvastī miracles (Buddha meditating beneath a blooming tree and the *nāga*) with *ye dhammā* formula. Ca. the second half of the 7th century CE. Ratchaburi province. Moulded tablet, clay; h. 14 × w. 11.5 cm. Ratchaburi National Museum (No. 242/2533). (**a**) Font Side; (**b**) back Side. After Bhumadhon and Phongpanit (2015 [2558 BE], fig. 10–8).

One fine example of a votive tablet belonging to the *yesaṃ* group, held today in the collection of Wat Matchimawat in Songkhla but probably originating from Thailand's central region (Figure 4a,b), is an inscription occupies five lines (instead of the usual four), with a slight deviation of the expected final syllables in lines 2, 3, and 4. The deviations are probably for reasons of equal distribution of *akṣaras* over lines (Revire 2014, p. 259).

(1)   *ye dhammā hetupprabhavā*
(2)   *yesaṃ hetuṃ tathāgato*
(3)   *āha tesañ ca yo niro-*
(4)   *–dho evaṃvādī mahāsama-*
(5)   *–no.*

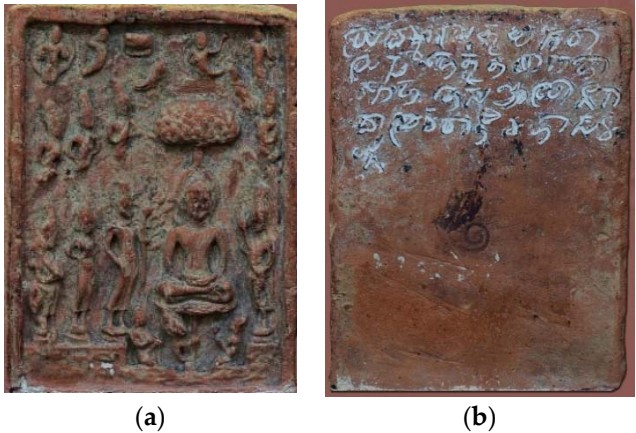

(**a**)                                    (**b**)

**Figure 4.** The Śrāvastī miracles (Buddha meditating beneath a blooming tree and the *nāga*) with *ye dhammā* formula. Ca. 7th–9th centuries CE. Moulded tablet, clay. Songkhla National Museum. (**a**) Font Side; (**b**) back Side. After Revire (2014, fig. 15a,b).

These three examples all belong to the *yesaṃ* group. The language used in a number of these inscriptions is not orthographically correct, reflecting the fact that the common tongue was extremely fluid and diverse (Saraya 1999, p. 196). They have in common certain spelling peculiarities in which *mahāsamano* is spelt with a dental *n* rather than the standard retroflex *ṇ* and *ti* is omitted; these are common to other occurrences of *ye dhammā* inscriptions belonging to this group and cannot, therefore, be simply deemed as a scribal error (Revire 2014, p. 259). The inscription of *ye dhammā* verse found in Nakhon Pathom follows the canonical group (*tesaṃ*), while the inscriptions from the *yesaṃ* group appear in Ratchaburi province, central Thailand, and are far more numerous and widespread. This indicates that "the two recensions also have some distinct geographical realities with the latter *yesaṃ* inscriptions marking the furthest extension of ancient Pāli literacy in northeast Thailand (Si Thep in Phetchabun Sema in Nakhon Ratchasima and Nong Bua Daeng in Chaiyaphum)" (Revire 2014, p. 259). Even though the sacred verses in these tablets are inscribed in both completed and truncated forms, they still embody the whole dharma.

The miracles of superpowers and the miracle of instruction are illustrated on these votive tablets, depicting the Śrāvastī miracles on the front along with the *gāthā* on either the front or back. In renderings of the Śrāvastī miracles on the Dvāravatī artefacts (see Sirisawad 2022, pp. 12–16), the Buddha is depicted in two types of gestures, either in meditative gesture (*dhyānamudrā*) or a gesture symbolising teaching or exposition (*vitarkamudrā*). The Buddha in either *dhyānamudrā* or *vitarkamudrā* is found on the artefacts without any verse, while on those which include the *ye dhamma*, he appears in *dhyānamudrā*. These gestures can be linked to the narrative elements of the Buddha's preaching of the dharma and to the miraculous demonstration. The Buddha performed the miracles as a method of converting others; therefore, this episode, in which the three miracles are included is appropriate for the early stage of the adoption of Buddhism in the Dvāravatī period. The Śrāvastī miracles are aimed at overcoming the pride of non-believers and to inspire those

who witness it in order to generate a faithful mind towards Buddhism. In the same way, the adoption of Buddhism in circa seventh century CE was a continuing process of integrating foreign culture and beliefs. Buddhism combined both traditional and foreign beliefs, placing them together within an integrated context. The depiction of the Śrāvastī miracles might cause indigenous people to adopt Buddhism amid such cultural diversity and local beliefs. According to the *Abhidharmakośabhāṣya*, a person who harbours a hostile, unbelieving, and non-zealous thought is made to produce the thought of refuge, faith, and practice by means of these three miracles (Abhidh-k-bh 7.47; La Vallée Poussin and Sangpo 2012, p. 2251). Additionally, these sacred verses, depicted in both the written textual tradition and on the artefacts, are perceived as the Buddha's miracle of instruction, which is higher than the other two miracles. Even some verses such as *ye dhammā* were not directly the word of the Buddha but of his disciples, they do cause the listener to be converted and to have faith in Buddhism. Through the miracle of exposition, persons are caused to produce salutary and agreeable effects; the preacher teaches, in truth, the means to salvation and happiness (Abhidh-k-bh 7.47; La Vallée Poussin and Sangpo 2012, p. 2252). Cœdès makes clear in his seminal article on the so-called "votive tablets" that the *ye dhammā gāthā* "must rapidly have acquired in the eyes of the ancient Buddhists a sort of magic virtue and may well have seemed to them a quite irresistible charm for the conversion to the faith of any who had not heard it" (Cœdès 1926, p. 6). In essence, the Śrāvastī miracles were an extremely complex and confusing legend that allowed the artist and his patron great latitude in the plastic depictions. "Unlike the other episodes of his life, the Śrāvastī episode is identified by not one symbol, a result, perhaps, of the unusually complex and diverse textual renderings" (Brown 1984, p. 29).

The cultic practice of ritually copying, and thereafter reciting, such sacred verses on artefacts made of clay, brick, stone, metal, and so on, was well established in South Asia and slightly later in Southeast Asia (Revire 2014, p. 260). It was presumably first employed with the making of Buddha images as described by Yijing, in the seventh century CE:

> Again, when the people make images and *caityas* which consist of gold, silver, copper, iron, earth, lacquer, bricks, and stone, or when they heap up the snowy sand (lit. sand-snow), they put in the images or *caityas* two kinds of *sarīras* [i.e., relics]. 1. The relics of the Great Teacher. 2. The *gāthā* of the Chain of Causation [i.e., *ye dhammā* or *paṭiccasamuppāda gāthās*]. [ . . . ] If we put these two in the images or *caityas*, the blessings derived from them are abundant. (Takakusu 1998, pp. 150–51).

Previous studies have shown that inscribing the *ye dharmā* verse and its installation in *stūpas*, images, and paintings has multiple functions. The sacred verses in which the very essence of the Buddha's teaching are summarized as a function of a "*Dharmaśarīra*" or "relic of the teaching" and they have been seen as closely connected to *stūpa* construction and the cult of relics, as mentioned in Yijing's work. The verses function as a representation of the Buddha himself. They are honoured and respected as if they were a relic of the Buddha (Revire 2014, p. 260). Skilling has proposed another function that is connected with the "consecration ritual" for the *gāthās* (Skilling 2008a, p. 251; Skilling 2008b, p. 507). Since these *gāthās* were accepted as a substitute for the Buddha, they could be inserted in *caityas*, *stūpas*, Buddha images, or other sacred items, at a particular spot "as part of a consecrating ceremony devised to empower the artefacts" and "would presumably have the effect of authoritatively legitimizing the object and that spot as a sacred and cultic center" (Revire 2014, p. 260). The Pāli *ye dharmā* verse found in the Dvāravatī culture, as well as other versions preserved in Prakrit and Sanskrit found elsewhere in Southeast Asia, "were deliberately chosen for their 'ritual' and 'cultic' nature, and thus also for their alleged supernatural power" (Revire 2014, p. 261). For reasons of language and paleography, it would have been uncommon that an ordinary person could decipher or engrave these verses and was therefore most likely conducted by ritual specialists, scholars, or learned monks. When the *gāthās* are used for such ritual purposes and inscribed on material objects, they "often no longer functioned as a means for communicating their verbal contents"

(Revire 2014, p. 260). These sacred verses have another important function in serving to generate religious merit (P. *puñña*; Skt. *puṇya*) for the sponsor or donor, the scribe who engraves them and the person who recites the *gāthās* (Revire 2014, p. 261). Although it cannot be said with certainty that they would have obtained the dhamma-eye, as in the period when the Buddha was alive, they could at least have derived great results from the meritorious act and from their belief in the Buddha. The *ye dhamma* stanza found in votive tablets is one piece of evidence that emphasizes the presence of the merit-making effort among Dvāravatī people.

*2.4. Śrāvastī Miracles as Supporting the Idea of Making the Buddha Images as an Act of Merit*

A number of scholars have also underlined that votive tablets were produced both as mementoes for pilgrims visiting the four important sites in Buddhism (the birth place of the Buddha, the place of the enlightenment, the place of the first sermon, and the place of the Buddha's parinirvāṇa) and in memorial of the Buddha (Buddha Monthon Construction Administration Committee 1988; Saraya 1999, p. 190).

Skilling argues that Buddhist sealings are not artefacts that emanate from the pilgrims' mementoes of Indian sites but are rather local ritual artefacts produced at a specific place and to be installed (not to be displayed, or even seen) in the base or relic-chamber of a *stūpa* or in caves: they are, namely, "witnesses of autonomous practices and of local ceremonies" (Skilling 2008a, pp. 248–49). An investment in ritual actions may eventually yield great results. The votive tablets have been produced in such great numbers, especially in Dvāravatī, as "products of a ritual ideology of mass production—the augmentation of merit by multiplication of images" (Skilling 2008a, p. 249). This ritual ideology spread across the Bay of Bengal. However, according to Ghosh (2014, p. 13), they could be both pilgrims' memento and products of a ritual ideology manufactured locally. An important phenomenon has been noticed by Ghosh that "pilgrims travelling to important religious centres carried home mementoes relating to a particular sacred space. Therefore, tablets could be a very good choice considering their size and portability. Specimens of such tablets could have been used as models for further replication or innovation in their own locality." (Ghosh 2017, p. 37).

The production of votive tablets in the Dvāravatī period were dedicated to the religion and were clearly the products of the ideology of merit[23], which is common in Indic religions. Regarding the reasons for producing votive tablets as an act of merit, Cœdès argues the following:

> But *phra phim* must have ceased at an early date to be regarded merely as souvenirs. With the development of a profound veneration for images, the act of making a statue of the Buddha or other figure symbolic of the religion had long been established as a source of merit. But to cast a bronze image or carve a statue of wood or stone was not within the reach of most people, and poor persons desirous of acquiring merit to assure their rebirth under more prosperous conditions, found in the impression of an effigy upon a lump of potter's clay, the means of accumulating such merit without the assistance of superior intelligence or wealth. Those having the desire and the leisure to do so, might make a very large number of such impressions (Cœdès 1926, p. 4).

Corresponding to the above statement, Skilling (2017, p. 25) states "Buddhas were produced to make merit for oneself and for others. The benefits of 'making images' are proclaimed in manuscripts and sermons." In the *Puññakiriyavatthusutta*, three foundations of meritorious actions have been identified: giving (*dāna*), virtue or morality (*sīla*), and mental development or meditation (*bhāvanā*) (AN iv 239; DN iii 218; It 51). The first, dāna, is of greatest relevance to the present research[24]. Revire (2014, p. 242) states that "the practice of *dāna*, however, has had considerable social and economic significance in all Buddhist cultures and has left a large corpus of archaeological material and inscriptions

for study." The production of any Buddhist artefact, image, or monument is based on the ideology of "Giving."

To be produced, all material objects must have patrons and donors as sponsors who thus obtain merit depending on the recipients of the donation. A study of inscriptions on Buddha statues, votive tablets, Sema stones, and other artefacts of the Dvāravatī culture in northeastern Thailand reveals the belief the laity held in fashioning Buddhist objects or establishing Buddhist sites for the purpose of cultivating merit (Dhamrungrueng 2015 [2558 BE], p. 84).

One ancient Mon inscription from Maha Sarakham province (eighth–ninth centuries CE.), written in Southern Brahmī script and inscribed on the reverse of baked, clay votive tablet that depicts the Śrāvastī miracles, reads: *naiavoapuṇya kamaraṯeṅ baiḍaka romārskuṅda sjāṯisamӑr* (Saraya 1999, p. 199)[25]. This was translated in Thai by the Fine Arts Department (กรมศิลปากร) as "บุญอันนี้ในกอมระตาญพร้อมไปด้วยสหายของตนผู้เป็นสามัญชนได้ร่วมกันสร้างไว้" [Kamaraṯen made this merit together with his friends, who are common people] (The Fine Arts Department 1986 [2529 BE], vol. 2, pp. 77–81; Saraya 1999, p. 199). Revire offered another possible translation of this inscription: "this is the work of merit of the king. May I not be born to a mean existence" (Revire 2014, p. 247) which does not correspond with Thai translation found in Uraisi Varasarin (1988 [2531 BE], pp. 505, 513): "นี้คืออบุญกำมุรเตงฝไทกโรมุ ขอจงได้สืบชาตินี้" (this meritorious act was made by the king. May I obtain (the benefit of making Buddha image) within ten lifespan) (as cited in Dhamrungrueng 2015 [2558 BE], p. 40). The Sanskrit word punya, which means "merit", appears on this votive tablet:[26]

(1)  *naiavoapuṇya* (ไนอํโวอํปุณย)

(2)  *kamaraṯeṅ baiḍa ka* (กมรเตง์ ไปฑ ก)

(3)  *romārskuṅda* (โรม์อาร์สกุ ฑ)

(4)  *sjāṯisamӑr* (ส์ชาติสมร)

Inscribing the donor's name and an aspiration on tablets does not seem to have been the usual custom in India, the Dvāravatī culture (the Chao Phraya valley), or the Malay Peninsula (Skilling 2008a, p. 252). There are two variations of reading of the Donative inscription on this Maha Sarakham tablet (Figure 5). Both mention the donor(s), who use of the Khmer royal title *kamaraṯeṅ baiḍa karom* (กมรเตง์ ไปฑ ก์โรม์) which appears here for the first time in Mon inscriptions (Bauer 1991, p. 46). This word is normally used with names, titles, and epithets of rulers or kings to indicate their social positions (The Fine Arts Department 1986 [2529 BE], vol. 2, pp. 77–81). It suggests the favour of the ruling elite towards merit making ceremonies which became popular among local Mon speaking people (Saraya 1999, pp. 167). Following the first suggested reading, the elite, the rulers, and the ordinary people, the ruled are the donors[27], while in the second alternative reading, only the elite is implied. Saraya states that "the Monarch or Raja held the highest position in the ruling class. Dvāravatī kings seem also to have been bound by other associated beliefs and rituals. Buddhist merit making, in particular, was perceived as conferring transcendental power upon a monarch" (Saraya 1999, p. 167). Thus, the elite were the leaders, and the ordinary people were the followers under the protection of the leader. Although the production of tablets was open to all, they are not "the poor person's road to heaven" because many were produced by kings, members of the court, and senior monks (Skilling 2008a, p. 249). Here, the artefact of the Śrāvastī miracles in the Dvāravatī period is one such example.

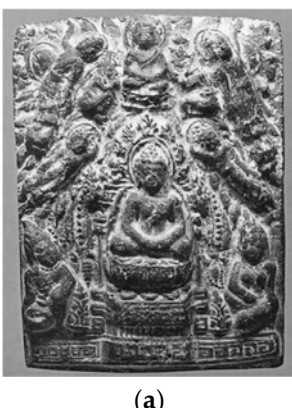 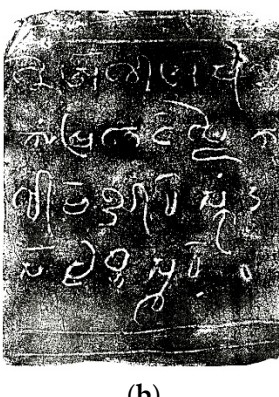

(**a**)　　　　　　　　　　　　　　　(**b**)

**Figure 5.** The Śrāvastī miracles (Buddha meditating on the lotus throne, mango tree, and multiplied Buddhas) with an ancient Mon with Khmer loan words inscription (Southern Brahmī script). Ca. the end of the 9th- early 10th centuries CE. Wat Non Sila Chumpae District; Khon kaen province. Moulded tablet, clay; h. 14 cm. Na Dun, Maha Sarakham prov. (Personal belonging). (**a**) Font Side; (**b**) Back Side. After Dhamrungrueng (Dhamrungrueng 2015 [2558 BE], fig. 14).

It is not evident from the short passage inscribed on the reverse of the votive tablet what the expected result of the donors in producing the Buddha images was, but it can be assumed from an alternative translation that the donors produced the sealings to either not be reborn in a lowly existence or to ensure a good "rebirth" in a future life. Corresponding to Skilling's study, from textual, epigraphic, and ritual evidence, especially from Siam and Burma, it suggests that the donors' primary inspiration for producing such artefacts was the quest for merit, liberation, and Buddhahood (Skilling 2008a, p. 248). This is the notion of "*ānisaṃsa*"[28] (in Thai อานิสงส์) as explained by Skilling as follows:

> The *ānisaṃsa* ideology has permeated the expression of Buddhism for centuries. It is entwined, interwoven, with the notions of merit (*puñña*, *puṇya*), giving (dāna), and aspiration (*adhiṭṭhāna*, the intentionality of action). *Ānisaṃsa* and a*dhiṭṭhāna* overlap with *dāna*: they are components of the dynamics of the interior world of *dāna*, they are its instrumentality (Skilling 2017, p. 2).

The intentionality of action or the expected result of the donor is not mentioned in votive tablets due to insufficient space. Only the donors' names can be inscribed in order that their meritorious act will be recognized and that they be rewarded with the goal of their intention (*adhiṭṭhāna*). Schopen mentions on the function of names in inscriptions that the donors "did not intend to leave a record, so it did not much matter whether it could be seen or read or understood . . . they wanted only, it seems, to leave their presence in proximity to another, more powerful presence . . . " (Schopen 2004, p. 392). Nonetheless, donors will certainly gain merit from the production of the Buddha images. For example, in some verses of the *Viriyapaṇḍita Jātaka* of the Burmese Fifty *Jātaka* collection, the Buddha mentions "happiness" as a primary goal in producing Buddha images:

> Therefore, a wise man, perceiving his own happiness, should always erect a Buddha image whether small or large. That image, well-made, either in wood, stone or in pure clay, or with sandalwood, with gold or silver, or with pearls or bronze. According to one's ability, an image of Buddha should be made. The donors, who constantly give, obtain happiness and wealth as long as they transmigrate in this world among gods or men . . . [29]

Another agency through which tablets found their home in distant lands was that of the traders. The inclusion of the traders relates to the strong connection between Buddhism and trade. Thus, tablets, such as amulets were voyaging objects and could be carried from one place to another either by pilgrims or by traders (Ghosh 2017, p. 37). Votive tablets are produced out of respect and faith in Buddhism. These auspicious objects carry with them

the values of remembrance and worship. Moreover, copying a Buddha image and verse was clearly perceived as a beneficial act of merit in itself, not only for the image maker or the scribe, but also for the donor who sponsored the act of copying. According to the Śrāvastī miracles' narratives, in which the Buddha duplicated himself to display the miracles, it is suitable for the image maker to create many Buddha images within one frame. As in Revire's statement: "this practice of duplicating a religious object was no longer motivated by the intention of preserving the 'exact word' or 'likeness' of the Buddha but rather was primarily aimed at merit-making" (Revire 2014, p. 262). Hence, producing and copying a large number of Buddha images, for example, miniature clay tablets or earthenware, and engraving them with such sacred verses selected from certain Buddhist texts are to necessitate great benefits and also as a means to pass along Buddhism from generation to generation.

### 3. Conclusions and Discussion

The analysis of the story of the Śrāvastī miracles, as gleaned from various Buddhist sources of different school affiliations and the artefacts and epigraphic records of Dvāravatī, has shown the significance of the Śrāvastī miracles in various ways. First, the Śrāvastī miracles serve as one of the Buddha's necessary deeds. Various Buddhist scriptures, exclusively in the Mūlasarvāstivāda versions, indicate that the miraculous demonstration at Śrāvastī is a unique and obligatory event that a Buddha must accomplish before he passes into final *nirvāṇa*. Second, the Śrāvastī miracles are principal miracles only performed by the Buddha. Various kinds of miracles are performed by the Buddha to overcome the pride of the *tīrthikas*. The accounts of the performance of the *yamakaprātihārya* or the "Twin Miracle" and the so-called "Great Miracle" or the *mahāprātihārya* can be performed only by a Buddha for the Theravādins while the Mūlasarvāstivādins records show only the Great Miracle is exclusively the act of a Buddha. Thirdly, the Śrāvastī miracles are a means of conversion to Buddhism. In the narrative, the three miracles, namely, the miraculous demonstration of supernatural powers, mind-reading, and expositions, are effective tools used by the Buddha to instruct and convert new followers and to bring people into the Buddhist fold in the same way previous Buddhas had done. The Buddha converts nonbelievers especially by means of the miracle of expositions, the best among the other two miracles. That is to say, he preaches the dharma either in verse or in prose in order to accommodate the multitude of living beings; they are either, in a discourse on the dharma that penetrates the four noble truths (*caturāryasatyasaṃprativedhakī*) and heedfulness (*apramāda*), or instruction on taking refuge in the Three Jewels. As a result of the combination of these miracles, the people obtain different effects.

The miracles of superpowers and the miracle of expositions are seen illustrated on Dvāravatī artefacts; for example, the buddha is in a meditative posture while performing the Great Miracle of duplicating himself, and at the same time, the central Buddhist credo, the *ye dharmā* verses, representing the Buddha's exposition, is found on the reverse side the tablet. The practice of inscribing and reciting such sacred verses on these artefacts has multiple functions; it is connected especially with the ritual and the cultic nature of generating religious merit. Moreover, the Śrāvastī miracles support the idea of making Buddha images as an act of merit. The Great Miracle depicting the multiple images of the Buddha supports the ritual ideology of mass production of Buddha images as a means of augmenting merit. Moreover, this study has clearly demonstrated that the Buddhist ideologies of gift giving and dedicating were essential to merit making. The ideology of Buddhist kingship also appears in the promotion of Buddhist practices such as merit making supporting the notion that the king as a patron of the religion. As such we see, in the inscription discussed above, that the king and his subjects were involved in an act of merit making. This has been recognized as "a means of accumulating charismatic power and served to build the faith of the laity. At the same time, it is a practice that was distinguished from traditional Buddhism adopted by the elite and ruling classes" (Saraya 1999, p. 219). These

significant features may explain the widespread adaption and the popularity of the Śrāvastī miracles' theme in Dvāravatī culture.

Various attempts have been made to demonstrate the early "establishment of Sri Lankan Buddhism" in Thailand during the so-called Dvāravatī period (Revire 2014, pp. 241–42) but there is little evidence. Buddhism in Dvāravatī constitutes a particularly cultural phenomenon in which Buddhism was recognized as an external belief system and was thence selected and adapted to play a new role in a local cultural system. This adoption of Buddhism was at the center of Dvāravatī culture. Although Buddhist practices and art are intimately related to Buddhism's arrival in the region, to the casual observer it can be difficult to make sense of this—often fragmentary—material culture, and the complex relationships between art, ideology, and rituals that are at its basis. Buddhist culture pervades Dvāravatī art, sculpture, and architecture. Additionally, most of Dvāravatī art was probably created to express narratives of the Buddha's lives such as the Śrāvastī miracles.

The traces that remain of Dvāravatī architecture, sculpture, and artefacts show that there is no single form of Dvāravatī Buddhism. It seems rather to be a combination of many schools of Buddhism, including, at least, the Theravāda and the Mūlasarvāstivāda schools. This observation is evidenced by the narratives of both schools being depicted in a number of artefacts illustrating the Śrāvastī miracles in stone, stucco, and terracotta from the central, northeastern, and southern parts of what is present day Thailand (see Sirisawad 2022). Nevertheless, the spread of Theravāda Buddhism through Pāli continued to exert a profound and continuous influence among the people until Buddhism became fully integrated into the culture. The act of merit making evidenced by some Buddhist artefacts shows that Buddhism had entered the lives of the general public. These practices spread along with the objects inscribed with the Pāli such as the *ye dhammā* formula discovered in present-day central Thailand. On the other hand, in the mid-seventh to mid-eighth centuries CE, the Mūlasarvāstivāda school was centered in northwest India at Mathura, in the Gandhāra region (present-day Pakistan and Afghanistan), and in Kashmir. According to the monk Yijing who was ordained in this school, t this tradition was widespread in southern China, Champa (present-day Vietnam) and the Indonesian Archipelago. It is thus reasonable to assume the Mūlasarvāstivādins were also active in what is now Thailand. Apart from votive tablets with the Pāli verses, Old Mon inscriptions were found in the contemporary northeast. Lying at the outskirts of the Dvāravatī cultural sphere, Sanskrit and Old Khmer are also attested in a few donation inscriptions. This study supports Revire's findings, in that "these linguistic trends may be an indication of the two major ethnic groups living in the region (i.e., Mon and Khmer) and of the sacred languages used (Pāli and Sanskrit)" (Revire 2014, p. 262).

A mixed character, tolerant of other faiths and of the local beliefs of indigenous people, was typical of Dvāravatī Buddhism. The interaction between Buddhism and local beliefs or faiths was not destructive in nature. On the contrary, they accumulated, were mixed, and then selected over a long period of time, and eventually were reflected distinctively in the unique style of Dvāravatī art, which extended across political or cultural borders as mentioned by Saraya: "although Dvāravatī Buddhism was not first time the faith had come into the region, it was as a cultural product, modified by local people (Saraya 1999, p. 218)". In this way, the religion was able to unify diverse groups of people, whereby both the elite and ordinary folk made merit. With the support and patronage of the ruling class, Buddhism absorbed a variety of traditional beliefs to form both the frame and core of Dvāravatī art, which encompassed varying Buddhist doctrines, Hindu motifs, and popular local traditions. Buddhism thus became deeply rooted in local communities, reaching out far and wide as communication routes expanded. This finding affirms Revire' s statement that "Buddhism firmly took root in Thailand only from this period onwards and not as far back to the time of King Asoka, circa 250 BCE, as often accounted in local traditions and school textbooks" (Revire 2014, p. 262). This popular form of Buddhism that emerged in Dvāravatī had been interpreted and transformed from its original form and was watered down to make it accessible to ordinary people.

**Funding:** This research received no external funding.

**Institutional Review Board Statement:** Not applicable.

**Informed Consent Statement:** Not applicable.

**Data Availability Statement:** Not applicable.

**Conflicts of Interest:** The author declares no conflict of interest.

## Abbreviation

| | |
|---|---|
| Abhidh-k-bh | Pradhan, P. (Ed.). (1967). *Abhidharmakośabhāṣyam of Vasubandhu.* (Tibetan Sanskrit Works Series 8). Patna. |
| AN | Morris, R., and Hardy, E. (Eds.). (1885–1900). *The Aṅguttara Nikāya* (Vols. 1–5). London: Pali Text Society. |
| Av-klp | Das, S. C., and Vidyābhūṣaṇa, Hari Mohan (Eds.). (1887). *Avadāna Kalpalatā* with its Tibetan version (Bibliotheca Indica; Collection of Oriental Works). Culcutta: Baptist Mission Press. |
| Avś | Speyer, J. S. (Ed.). (1958 [1902–1909]). *Avadānaçataka: A Century of Edifying Tales Belonging to the Hīnayāna.* The Hague: Mouton & Co. |
| BHSD | Edgerton, Franklin. (1953). *Buddhist Hybrid Sanskrit Grammar and Dictionary*, vol. 2: Dictionary. New Haven: Yale University Press. |
| cf. | Confer |
| Dhp | von Hinüber, Oskar, and Norman, K. R. (Eds.). (1995). *Dhammapada* with a complete Word Index compiled by Shoko Tabata and Tetsuya Tabata. Oxford: Pali Text Society. |
| Dhp-a | Norman, H. C. (Ed.). (1906–1914). *Dhammapadaṭṭhakathā* (Vols. 1–5). London: Pali Text Society. |
| Divy | Cowell, E. B., and Neil, Robert A. (Eds). (1987). *The Divyāvadāna: A Collection of Early Buddhist Legends now first edited from the Nepalese Sanskrit Mss. in Cambridge and Paris.* Delhi: Indological Book House. |
| DN | Rhys Davids, T. W., and Carpenter, Joseph E. (Eds). (1890–1911). *The Dīgha Nikāya* (Vols. 1–3). London: Pali Text Society. |
| DPPN | Malalasekera, G. P. (1937–1938). *Dictionary of Pāli Proper Names* (Vols. 1–2). London: J. Murray. |
| Fig. | Figure (pl. Figs.) |
| GM | Dutt, Nalinaksha (Ed.). (1939–1959). *Gilgit Manuscripts* (Vols. 1–4). Srinagar: Calcutta Oriental Press. |
| It | Windisch, Ernst (Ed.). (1889). *Itivuttaka.* London: Pali Text Society. |
| Ja | Fausbøll, M. V. (Ed) (1877–1896). *Jātaka*, together with its Commentary being tales of the anterior births of Gotama Buddha (Vols. 1–7). London: Trübner and Co. |
| Kv | Taylor, A.C. (Ed.). (1894–1897). *Kathāvatthu.* London: Pali Text Society. |
| LV | Vaidya, Paraśurama L. (Ed.). (1958). *Lalitavistara.* Darbhanga: Mithila Institute. |
| Mil | Trenckner, V. (Ed.). (1997 [1880]). The *Milindapañho: Being Dialogues Between King Milinda and the Buddhist Sage Nāgasena.* Oxford: Pali Text Society. |
| MPrS | the Gilgit manuscript of the *Mahāprātihāryasūtra*; Ed. and transl. in Sirisawad 2019 |
| Mp-ṭ | Sāriputta. (1961). *Sāratthamañjūsā [Manorathapūraṇī-ṭīkā]* (Vols. 1–3). Rankun. |
| MSV | the Mūlasarvāstivāda *Vinaya* |
| MSV-C | The parallel versions of the *Mahāpratihāryasūtra* from the Chinese Translation of the Mūlasarvāstivāda *Vinaya* (*Gēnběn shuōyíqièyǒubù Pínàiyē Záshì* 根本說一切有部毘奈耶雜事 (translated by Yijing 義淨, 710 CE), T. 1451 vol. 24, 207a–414b. |
| MSV-T | The parallel versions of the *Mahāpratihāryasūtra* from the Tibetan Translation of the Mūlasarvāstivāda *Vinaya* (*'Dul ba phran tshegs kyi gzhi*) (translated by Vidyākaraprabha, Dharmaśrīprabha and dPal 'byor, 9th century CE); Ed. and transl. in Sirisawad 2019 |
| Mvu | Senart, Émile. (Ed.). (1882–1897). *Le Mahāvastu: Texte Sanscrit publié pour la premiere fois et accompagné d'introductions et d'un commentaire* (Vols. 1–3). Paris: Imprimerie nationale. |
| Mvy | Ishihama, Y, and Fukuda, Y. (1989). *A New Critical Edition of the Mahāvyutpatti.* Studia Tibetica 16. |
| MW | Monier-Williams, M. (2002 [1872]). *A Sanscrit-English Dictionary.* Delhi: Motilal Banarsidass. |
| p. | page (pl. pp.) |
| Paṭis | Taylor, A. C. (Ed.). (1905–1907). *Paṭisambhidāmagga* (Vols. 1–2). London: Pali Text Society. |
| PrS(Divy) | the Prātihāryasūtra of the Divyāvadāna, ed. E. B.Cowell and R. A. Neil → Divy. |
| Ps-pṭ | Dhammapāla. *Līnatthapakāsinī* II. *Papañcasūdanī-purāṇaṭīkā.* |
| PTSD | Rhys Davids, T. W., and Stede, W. (Ed.). (1921–1925). *The Pali Text Society's Pali-English Dictionary.* London |
| Q | Peking xylograph Kanjur-Tanjur, Qianlong edition |

| Skt. | Sanskrit |
|---|---|
| SN | Feer, L. (Ed.). (1884–1898). *The Saṃyutta-Nikāya*. London: Pali Text Society. |
| Spk-pṭ | Dhammapāla. *Sāratthapakāsinī-purāṇatīkā. Līnatthapakāsinī* III. |
| Sv | Rhys Davids, T. W., Carpenter J. Estlin and Stede, W. (Eds.). (1886–1932). *Sumaṅgalavilāsinī, Buddhaghosa's Commentary on the Dīghanikāya* (Vols. 1–3).London: Pali Text Society. |
| Sv-pṭ | Dhammapāla. (1970). *Sumaṅgalavilāsinīpurāṇaṭīkā. Līnatthapakāsinī* I, edited by Lily de Silva. (Vols. 1-3). London: Pali Text Society. |
| SWTF | Bechert, Heinz. (Ed.). (1973). *Sanskrit-Wörterbuch der buddhistischen Texte aus den Turfan-Funden*, Begonnen von Ernst Waldschmidt. Göttingen. |
| T. | Junjirō Takakusu 高楠順次郎, Kaigyoku Watanabe 渡邊海旭, and Genmyō Ono 小野玄妙. (Eds.). (1924–1932). *Taishō shinshū Daizōkyō* 大正新修大藏經 [*Taishō Edition Tripiṭaka*]. 100 vols. Tokyo: Taishō Issaikyō Kankōkai. Online version available in: CBETA (Chinese Buddhist Electronic Text Association), http://www.cbeta.org/. |
| Th | Oldenberg, Hermann, and Pischel, Richard. (Eds.). (1883). *Theragāthā*. London: Pali Text Society. |
| Tib. | Tibetan language |
| Transl. | Translation |
| Upāyikā-ṭīkā | Śamathadeva's the *Abhidharmakośopāyikā-ṭīkā* (*Chos mngon pa'i mdzod kyi 'grel bshad nye bar mkho ba zhes bya ba*); Ed. and transl. in Sirisawad 2019 |
| Uv | Bernhard, Franz. (Ed.). (1965–1968). *Udānavarga*. Abhandlungen der Akademie der Wissenschaften in Gottingen, philologisch-historische Klasse, dritte Folge, Nr. 54 / Sanskrittexte aus den Turfanfunden, X. Gottingen: Vandenhoeck und Ruprecht. |
| UvViv | Balk, Michael. (1984). *Prajñāvarman's Udānavargavivaraṇa* (Vols. 1–2). Bonn. |
| Vin | Oldenberg, Hermann. (Ed.). (1879–1883). *Vinayapiṭaka.* (Vols. 1–5). London: Pali Text Society. |

## Notes

[1] For the edition of the text, see (Sirisawad 2019, pp. 17–51, 192–198).

[2] For the edition of the text, see (Falk and Steinbrückner 2020, pp. 3–42).

[3] The Chinese retains a version slightly different from the Tibetan in this fifth duty: "The fifth is to deliver all living beings who have received teachings only from the Buddha toward emancipation" (五者但是因佛受化衆生悉皆度脱) (Rhi 1991, p. 273).

[4] There are two lists of five essential deeds found in T. 125 [a] 622c12–15 (Bareau 1995, p. 200) and T. 125 [b] 703b17–20 (Rhi 1991, p. 21, note 36) and see the discussion in (Rhi 1991, p. 21, note 36).

[5] *Bhaiṣajyavastu*: Gilgit version (199v1: GM iii 1, 162.17; Clarke 2014, p. 90); the Tibetan version (Q1030, vol. 41, 260b4). For the Tibetan text and French translation see (Hofinger 1982, vol. 1, pp. 7–8) (Introduction), p. 33 (Text), pp. 175–77 (Transl.)

[6] The characteristic of the *iddhipāṭihāriya* is explained in the *Paṭisambhidāmagga* (i 111).

[7] For an analysis of these various terms, see (Fiordalis 2008, p. 47ff).

[8] The verb, *paṭi+√hṛ*, is found in Pāli texts in the sense of striking in return or against, while the form, *paṭi+ā+√hṛ*, is used more in the sense of taking away. See PTSD entries under *paṭiharati*, *paccāharati* and *harati*.

[9] The same explanation is at Ps-pṭ i 24, Spk-pṭ i 21, and Mp-ṭ i 24.

[10] The same explanation is found in DN iii 3; SN iv 290; AN i 170, v 327; Paṭis ii 227. The Sanskrit reads: *trīṇi prātihāryāṇi ṛddhiprātihāryam ādeśanāprātihāryam anuśāsanīprātihāryam*, see BHSD 392; SWTF III 229–230; Mvy 232–234; Mvu i 238, iii 137 (*dharmadeśanā*-instead of *ādesanā-pāṭihāriyaṃ*).

[11] Cf. the Dharmaguptaka *Vinaya* (T. 1428: 797a) and the *Dirghāgama* (T. 1: 9c–10a, 101c–102a).

[12] These miracles are commonly retained in the literature of several sects, such as the Theravāda, (Mūla)Sarvāstivāda, and Mahāsāṅghika, only with slight differences in phrasing, see (Rhi 1991, p. 30, note 62). For the Theravādins, it corresponds to the three kinds of "wonders" in the Blessed One's teaching, which are mentioned in various texts of the Pāli canon, such as the stories of the conversion of the Kāśyapa brothers at Urubilvā, as it appears in the *Mahāvagga* of the Pāli *Vinaya*, and in the Vinayas of other early Buddhist schools, see (Fiordalis 2008, pp. 73–86).

[13] Apart from the above occurrences in the versions of the Mūlasarvāstivādins, they also appear in the *Sahasodgatāvadāna* (*Divyāvadāna* no. 21), the *Rudrāyaṇāvadāna* (*Divyāvadāna* no. 37), *Saṅghabhedavastu* of the *Vinayavastu*, the *Prātimokṣasūtra*, the Mūlasarvāstivāda monastic code (see Skilling 1999a, pp. 441, 444, notes 8–10), and the *Kapphiṇāvadāna* of the *Avadānaśataka* no. 88 (Avś ii 105). Dealing with heedfulness, these verses are included in a great collection of verses, the *Udānavarga*, which is roughly comparable to the Theravādin *Dhammapada* and *Udāna* combined, in the "Chapter on Heedfulness" (*Apramādavarga*) (Uv iv 37–38). The two verses occur three times in the Pāli canon (see Skilling 1999a, pp. 442, 444, notes 11–12), in the *Aruṇavatisutta* (*Sagāthavagga, Saṃyuttanikāya*) (SN i 155–157), *Abhibhūtatheragāthā* (*Tikanipāta, Theragāthā* verses 255–257) (Th 31), and *Kathāvatthu* (Kv 203). Sometimes either the first or second occurs alone. The second part of the verse occurs in the *Mahāparinibbānasutta* (DN ii 121), and the first part occurs in the *Milindapañha* (Mil 244–245). Moreover, the verses exist in the *Nibbānasutta* (see Hallisey 1993, pp. 97–130), an allegedly non-canonical *sutta* of the Theravadin corpus whose Pāli witness probably derives from Southeast

Asia. The verses occur in the literature of other schools also. They are included in the "Chapter on Heedfulness" (*Apramāda*) in the "Gāndhārī *Dharmapada*", which belongs to the Dharmaguptaka school (see Skilling 1999a, p. 441), in Chapter 22 of the *Book of Zambasta,* an early work in Khotanese, and in the *Mahāsamājasūtra* (Chinese *Dīrghāgama, Sūtra* 19), which belongs to the Dharmaguptaka school (See Waldschmidt [1932] 1979, p. 194; Ichimura 2016, p. 139). They are cited by Bhavya, who was a leading exponent of the Madhyamaka (ca. 500–570 CE), in Chapter 3 of his *Madhyamakaratnapradīpa*, (see Skilling 1999a, pp. 442–43, 444, notes 13–17), in the *Bhadrapālaśreṣṭiparipṛcchā* (*Tshong dpon bzang skyong gis shus pa*, no. 39) of the *Mahāratnakūṭa* collection (Q 760, vol. 24, 'i 73b3–4). The *ārabhadhvaṃ niṣkrāmata* verses are engraved on a votive inscription at Nālandā dated to the reign of Mahendrapāladeva, a famous Pāla king of the late 9th century CE., wherein they are themselves described as a "*caitya* of the Blessed One, the Sugata (see Sastri 1999, pp. 106–107). In the Tibetan tradition the verses are inscribed in monastery vestibules or on cloth-paintings (*thangka*) depicting the "wheel of life" (*Bhavacakra*) or the Buddha (Ajanta cave 17 has a depiction of the *Saṃsāracakra*, see (Schlingloff 2013, vol. 3: XVII, 20)). The verses are instructive examples, and they are put to difference purposes in different written and visual media.

14  These verses are included in the *Yugavarga* of the *Udānavarga* (Uv xxix 1–2) and the *Jaccandhavagga* in the Pāli *Udāna* (Ud vi 10).

15  See also (Skilling 1997, vol. 1, pp. 306–8, vol. 2, pp. 464–67). Furthermore, neither the Sanskrit *Dhvajāgra-sūtra* from Central Asia nor the Chinese versions include the verses (Skilling 1991, p. 240).

16  Rotman (2008, p. 433, note 628) reads *buddhakārakṛtāḥ.*

17  It could be read as *alpa* or *alpān* cf. Tib. *cung zad, chung ba.* However, Hirakawa's (1973–1978) Index to *Abhidharmakóśabhāṣya* suggested '*lpān.*

18  The four stages of penetrating insight (*nirvedhabhāgīya*) are the four stages on the path of application (*prayogamārga*): heat (*uṣmagata*), tolerance (*kṣānti*), summit (*mūrdha*), and the highest worldly dharma (*laukikāgradharma*). The first three are themselves sub-divided into three degrees—weak, medium, and strong—hence there are ten stages in all. (Rotman 2008, p. 452).

19  A following episode concerns Upatissa (Sāriputta) and Kolita (Mogallāna), two young Brahmins and pupils of the recently deceased Sañjaya, who became the guardians of the group [of followers of Sañjaya], the managers of the group. One day Upatissa encountered the Buddha's disciple Assaji, who was out for alms. Upatissa, impressed by the Assaji's deportment, asked him, 'Who, O monk, is your teacher, or: under whose guidance did you enter the religious life, or: whose dharma do you declare?" Assaji answered that 'the ascetic Gautama, a son of the Śākyas, . . . is my teacher. Under his guidance I entered the religious life. I declare his dharma.' This leads to the conversion of not only both Upatissa and his companion Kolita but also of their followers (Vin i 39ff).

20  See the database of artefacts Type 4 in Sirisawad (2022), Figure 4.

21  A double pa and a subscript ra seem to appear, even though the reading on the tablets is rather obscure, making the compound -*ppra* in *hetupprabhavā* clear.

22  See the database of artefacts Type 6 in Sirisawad (2022), Figure 6.

23  See the list of Buddha images, miniature tablets or shrines, or the sponsorship of Buddhist buildings in (Revire 2014, pp. 250–52, table 2).

24  Giving (*dāna*) is one of the essential preliminary steps of Buddhist practice. The act of giving is also emphasised in ancient Buddhist stories and tales such as *Jātakas* and *Avādanas*. It is the supreme virtue perfected by all Bodhisattvas in their long path toward perfection (*pāramitā*) and the perfect self-enlightenment (*sammāsambodhi*) (Revire 2014, p. 243). Naturally, the average layperson is not expected to make so great a sacrifice as Bodhisattvas did. For most people, the practice of *dāna* is limited to material support in order to make merit.

25  Bauer (1991, p. 42) gives another transliteration as "*nai vo' puṇya kamrateṅ pdai karom'* or *sku⌣ das jāti smar*". He also argues that the term *nai* for "this" may be a variant of a similar Khmer word which would suggest contact with Khmer populations in this region.

26  There are variant spellings of this word such as *puñ, piñ, pinna, puṇya,* or *puṇa.* A few significant examples are given in (Revire 2014, pp. 244–47).

27  Several Mon inscriptions mention communal merit making by the assembled elite, the middle-class people and the general populace, see the (The Fine Arts Department 1986 [2529 BE], vol. 2, pp. 60–66, 71, 81, 103; Dhamrungrueng 2015 [2558 BE], pp. 84–89).

28  *Ānisaṃsa* can mean both a promise of benefit and reward and the benefits and reward themselves. For further discussion on the range of the term, see (Skilling 2017, pp. 5–6).

29  *Viriyapaṇḍitajātaka*, in *Paññāsa-Jātaka* Jaini (1983, vol. 1, No. 25, pp. 297–308); (Jaini 1986, vol. 1, pp. 306–16). The verses use the terms *Buddha-bimba, Buddha-paṭimā,* and *Buddha-rūpa.* For other examples of benefits from making Buddha images, see (Skilling 2017, pp. 26–28).

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
