# Peer review of "Significance of the Śrāvastī Miracles According to Buddhist Texts and Dvāravatī Artefacts"

_religions, doi:10.3390/rel13121201_

Round 1
Reviewer 1 Report
This is a very well written and researched article, and it was a pleasure to read. My main recommendation is that you introduce to the reader “Dvāravatī culture” at the beginning of the essay to better contextualize it. Many readers will not be familiar with Thai cultural history and will thus need this. I also found the following minor problems which you might want to consider revising:
Line 22: Please identify “Dvāravatī culture” for the reader.
Line 35: Should the title “kṣudrakavastu” be capitalized like the other titles?”
Line 45: The text “or [=Av-klp]which” is unclear. Should this be emended to “[=Av-klp], which”?
Line 75: The text “was in a constant integration” seems awkward. Do you mean “was in the process of being integrated”?
Lines 75-79: The following sentence is problematic, “What might happen is that, the promulgators of Buddhism at that time, supported by the ideology of merit, find the narrative of Śrāvatī Miracles could serve as an important means to glorify the Buddha and shatter indigenous or other religious belief systems in order to make way for Buddhism.”
I’d recommend revising it as follows: “The promulgators of Buddhism at that time, supported by the ideology of merit, seem to have found that the Śrāvatī Miracles narrative could serve as an important means to glorify the Buddha and shatter indigenous or other religious belief systems in order to make way for Buddhism.”
Line 84: Correct “have” to “has”
Line 87: Correct “2.1Ś. rāvastī” to “2.1. Śrāvastī”
Line 93: Correct “¬kṣudrakavastu” to “Kṣudrakavastu”?
Line 144: Correct “2.2Ś. rāvastī” to “2.2. Śrāvastī”
Line 175: Correct “2.3Ś. rāvastī” to “2.3. Śrāvastī”
Line 275: By “commended” do you mean “commanded” or “recommended”?
Line 352: Correct “artifects” to “artefacts”
Line 355: Correct “artifects” to “artefacts”
Line 364: Correct “¬kṣudrakavastu” to “Kṣudrakavastu”?
Line 453: Correct “This” to “These”
Line 633: Correct “akṣaras” to “the akṣara” or “the syllable”
Line 770: Correct “2.4Ś. rāvastī” to “2.4. Śrāvastī”
Author Response
Dear Reviewer,
Thank you so much for taking the time to review my article as well as for providing me with valuable feedback.
I have accepted and acted on all your suggestions. In particular, I have also added a few lines on Dvāravatī culture as suggested by you.
I attached here a table detailing, point by point, your suggestions/feedback, and the actions taken by me.
I hope I have revised the manuscript to your satisfaction.
Thank you again for your time and effort.
With regards,

Reviewer 2 Report
This is a good study of the subject matter. Typographical errors do not indicate a lack of competence in handling the literature on the part of the authors.
Sometimes there is a strange symbol ¬.
Line 65 Śrāvatī Miracles. The Śrāvati Miracles ...
Line 87 and passim. 2.1Ś. rāvastī
Line 93 MSV-T: des par mdzad>nges par mdzad
Line 107 Bhaisajyavastu>Bhaiṣajyavastu
Gāyaśirṣa>Gayāśirṣa
acchariyaabbhuta>acchariya-abbhuta
Buddhaʼs lu- 320 (the mark ' should be unified)
Lines 420-422:
He teaches that the three realms are under the rule of the three principles: 419
“These three realms (三界) are impermanent (無常) and nonexistent (無). There is no-self 420
(無我). [Existence] is suffering and emptiness (苦空). Nirvāṇa (滅), unconditioned (無爲), 421
brings peace (安)” (T. 193: 86c13–17). 422
☞The Buddha would not have said that "These three realms are" "nonexistent (無)." The text (T. 193: 86c16) has 無堅 unsubstantial instead of 無 nonexistent. The original Sanskrit must be asāra(ka), unsubstantial.
’dus byas 487
dang ’dus ma byas rnam par nyes pa
> ... rnam par nges pa
Footnote 6
6 It should read alpa or alpān cf. Tib. cung zad, chung ba.
☞Hirakawa Index has already suggested a correction, 'lpān.
ānāgāmiphala>anāgāmiphala
Author Response
Dear Reviewer,
Thank you so much for taking the time to review my article as well as for providing me with valuable feedback.
I have accepted and acted on all your suggestions. I attached here a table detailing, point by point, your suggestions/feedback, and the actions that I have taken.
I hope I have revised the manuscript to your satisfaction.
Once again, thank you very much.
With regards,
